# WHY SGD IS NOT BROWNIAN MOTION: A NEW PERSPECTIVE ON STOCHASTIC DYNAMICS

## ABSTRACT

The conventional wisdom in deep learning theory often models Stochastic Gradient Descent (SGD) as a Brownian particle, described by a Langevin equation. In this work, we challenge this paradigm and propose a more fundamental perspective: SGD is best understood as deterministic dynamics within a fluctuating loss landscape. From first principles, we derive a master equation for the parameter evolution and its corresponding Fokker-Planck equation, which exhibits differences from the standard form used for Brownian motion. We analyze the resulting dynamics near minima, where the loss is approximately quadratic, and identify distinct behavioral regimes. The most intriguing behavior emerges in the presence of valleys. We demonstrate that in this regime, the dynamics do not converge to a stationary distribution. Instead, individual SGD trajectories diffuse along the floor of these valleys with an effective diffusion coefficient proportional to the learning rate. We empirically validate these theoretical claims through experiments on deep learning tasks in computer vision and natural language processing.

## 1 INTRODUCTION

Stochastic Gradient Descent (SGD) has been the cornerstone of neural network training since the earliest days of deep learning (Rumelhart et al., 1986). Despite the rapid evolution of architectures, datasets, and computational resources, SGD continues to play a central role in training modern over-parameterized models (Bottou, 2010; Goodfellow, 2016) such as Transformers (Vaswani et al., 2017) and large-scale language models (Team et al., 2025; Hernández-Cano et al., 2025). Its simplicity, scalability, and empirical success make it a robust baseline against which more sophisticated optimizers are compared (Semenov et al., 2025; Wen et al., 2025). All the most powerful modern optimization methods, such as Adam (Kingma, 2014) or the recently introduced Muon (Jordan et al., 2024), are fundamentally based on SGD Bernstein & Newhouse (2024). Remarkably, even in contemporary benchmarks, recent work (Srećković et al., 2025) has shown that for sufficiently small batch sizes, vanilla SGD can perform nearly on par with Adam, highlighting that sophisticated optimizers may effectively reduce to plain SGD in this regime. These findings emphasize that, decades after its original formulation, SGD is not only historically foundational but also practically indispensable in modern machine learning.

Despite its widespread adoption, a complete theoretical understanding of SGD dynamics is still lacking. A common line of thought interprets SGD through the lens of statistical physics, modeling it as a noisy relaxation process akin to Langevin dynamics (Mandt et al., 2017; Simsekli et al., 2019; Xie et al., 2021). This framework suggests that stochastic updates can be viewed as perturbations around a smooth loss landscape, leading—under simplifying assumptions—to a stationary Gibbs distribution. While this picture is conceptually appealing, it rests on approximations such as treating gradient noise as Gaussian via the Central Limit Theorem . As we discuss in the related work (Section 2), these assumptions can break down in practical deep learning scenarios, and recent findings point to systematic departures from the Brownian motion paradigm.

A crucial step toward exposing such limitations came from the study of (Feng & Tu, 2021a). By analyzing SGD trajectories near solutions, they established a robust and universal *inverse variance–flatness relation*, showing that flatter directions in the loss landscape correspond to smaller fluctuations.

Building on these insights, our theoretical framework begins by revisiting the assumptions behind existing interpretations of SGD dynamics. While analogies to physical systems have provided useful intuition, they overlook key properties of SGD that stem from its algorithmic origin. In particular, traditional theoretical approaches have largely borrowed concepts from statistical physics, treating SGD noise as analogous to Brownian fluctuations in a thermal bath. However, this analogy is limited because, unlike physical systems where noise sources are external and uncontrollable, SGD's stochasticity arises purely from mini-batch sampling. This intrinsic difference means SGD noise is structure-dependent and fundamentally distinct from Gaussian noise. We address this gap by proposing a novel perspective:

**SGD is NOT Brownian motion, but deterministic dynamics in a fluctuating potential.**

To summarize, our main contributions are as follows:

- **Discrete Fokker–Planck framework.** Starting from the exact SGD update, we derive a discrete Fokker–Planck equation and compare it with the commonly used Langevin-based approach. We demonstrate how an imprecise accounting of terms with different orders of the learning rate causes important terms to be left out of the resulting Fokker-Planck equation.

- **Analysis near critical points.** We conduct a detailed analysis of SGD dynamics in the vicinity of a critical point of the loss function, where the landscape can be approximated by a quadratic form. Within this framework, we derive a general expression for the variance of parameter trajectories in terms of the statistics of the Hessian.

- **Unified theoretical picture of SGD dynamics.** Our framework organizes SGD behavior in the Hessian eigenbasis. Along directions with strictly positive curvature, parameter fluctuations saturate in accordance with the inverse Einstein relation (Feng & Tu, 2021a). In nearly-flat or weakly unstable directions, no stationary distribution emerges, therefore individual trajectories drift apart and fail to represent the ensemble. Finally, in valley-like regions, SGD realizes an effective diffusion process with a coefficient proportional to the learning rate.

- **Empirical validation across domains.** We corroborate these theoretical predictions with controlled experiments on vision and language benchmarks. By analyzing Hessian spectra, variance dynamics along eigendirections, and the distribution of converged losses, we validate that the observed behavior of SGD is consistent with our theoretical predictions.

## 2 RELATED WORK

**Langevin-based views of SGD.** A classical line of research models SGD as a discretization of Langevin dynamics, establishing links to equilibrium statistical physics. In the work (Mandt et al., 2017) the authors argued that under sufficiently small learning rates, SGD can be interpreted as approximate posterior sampling around local minima, with minibatch noise acting as an effective temperature. In (Smith & Le, 2017), the relation between gradient noise scale, batch size, and learning rate was analyzed, leading to the so-called "noise-scale" rule. Within a such framework, SGD has been associated with implicit regularization mechanisms and a preference for flat minima (Hu et al., 2017; Chaudhari & Soatto, 2018; Zhu et al., 2019; Yang et al., 2023). Nevertheless, these descriptions crucially rely on Gaussian approximations and stationarity assumptions in the long-time limit.

**Heavy-tailed and non-Gaussian noise.** Moving beyond Gaussian assumptions, a series of works have examined the distributional properties of minibatch gradients. In (Simsekli et al., 2019), the authors provided empirical evidence that minibatch noise displays heavy-tailed behavior, better captured by $\alpha$-stable laws rather than Gaussian models. Building on this observation, the studies (Simsekli et al., 2020; Gurbuzbalaban et al., 2021) investigated the implications of heavy-tailed noise, showing that it can accelerate exploration in non-convex landscapes and lead to diffusion behaviors qualitatively distinct from those induced by Brownian motion. Further insights were presented in (Nguyen et al., 2019), where the first exit time of SGD from sharp minima under heavy-tailed noise was theoretically characterized. Collectively, these results challenge the validity of Gaussian-based Langevin approximations.

**Anisotropy and structure of gradient noise.** Another line of inquiry has highlighted the anisotropic character of minibatch noise. In (Jastrzębski et al., 2018), the covariance structure of gradient fluctuations was shown to align with the Hessian spectrum, implying stability properties that depend on direction. This perspective suggests that SGD noise cannot be reduced to isotropic Brownian forcing; rather, its interaction with curvature determines the effective dynamics. Furthermore, in (Zhu et al., 2019) the authors emphasizes that the anisotropic nature of SGD is a crucial aspect in studying the escaping from sharp minima. The specific noise structure induced by SGD has been shown to guide the optimization trajectory towards regions of the loss landscape characterized by flat minima, which typically exhibit better generalization.

**Alternative dynamical formulations.** In parallel, other perspectives have sought to characterize SGD dynamics in ways that go beyond the Langevin analogy. In the work (Yaida, 2020) it was shown that finite-width neural networks exhibit fundamentally non-Gaussian fluctuations, providing a different lens on stochastic dynamics in training. A complementary viewpoint was proposed in (Feng & Tu, 2021b), where the learning dynamics of neural networks were analyzed as distinct phases, depending on data quality and optimization noise. Together, these studies emphasize that SGD operates in intrinsically non-equilibrium regimes not captured by classical Langevin formulations. Our work builds on this direction by deriving stochastic dynamics from first principles and by validating these predictions empirically.

## 3 DYNAMICS ON THE RANDOM LANDSCAPE

**Brownian motion vs. motion in a random potential.** Consider SGD with replacement, where each step is driven by the gradient of a randomly sampled minibatch loss $L_n(\vec{w}_n)$. This means that at each iteration the minibatch is drawn independently from the full dataset. The basic update rule for the network parameters $\vec{w}_n$ reads

$$\vec{w}_{n+1} = \vec{w}_n - \eta \vec{\nabla} L_n(\vec{w}_n),$$

where $\eta$ denotes the learning rate.

A natural analogy is to view this as Brownian motion, i.e. the trajectory of a particle influenced by random forces in a static potential (Fig. 1, left). This perspective leads to the Langevin-like formulation

$$\vec{w}_{n+1} = \vec{w}_n - \eta \vec{\nabla} L(\vec{w}_n) - \sqrt{\eta} \, \vec{\xi}_n,$$

where $\vec{\xi}_n$ is interpreted as a stochastic force. A critical aspect of this formulation is that the scaling of the noise term $\sqrt{\eta} \, \vec{\xi}_n$ with the learning rate $\eta$ is chosen specifically to ensure that the derived Fokker–Planck equation possesses a well-defined continuous limit (see Appendix A). Currently, the application of Langevin dynamics to model SGD is a topic of active research, with a substantial and growing body of literature dedicated to it (Ustimenko & Beznosikov, 2024; Dalalyan, 2017; Raginsky et al., 2017; Durmus & Moulines, 2019; Li et al., 2019; Xie et al., 2021). Nevertheless, this construction does not faithfully reproduce the actual SGD update, in which the stochastic contribution naturally enters at order $\eta$, but not $\sqrt{\eta}$. As a consequence, the Langevin-based view misses essential terms in the probability evolution and should be regarded as an approximation rather than the true process (see Appendix A).

Observing SGD more carefully suggests a different physical picture. The randomness does not appear as an external force acting on a fixed potential, but rather through the fact that the potential itself changes from step to step due to minibatch sampling. In other words, SGD corresponds more closely to the deterministic motion of a particle in a fluctuating potential, analogous to the dynamics of a passive scalar advected by turbulent flow (Fig. 1, right) Klyatskin (2010). Unlike Brownian motion in physics, where noise sources are external and uncontrolled, in SGD the source of randomness is fully algorithmic and transparent. This distinction enables us to analyze SGD noise from first principles rather than relying on phenomenological analogies.

Within the paradigm of a fluctuating potential, it is natural to decompose the minibatch loss into its deterministic and stochastic components:

$$L_n(\vec{w}_n) = \bar{L}(\vec{w}_n) + \delta L_n(\vec{w}_n), \tag{1}$$

where $\bar{L}(\vec{w}_n)$ denotes the mean loss and $\delta L_n(\vec{w}_n)$ represents minibatch-induced fluctuations.

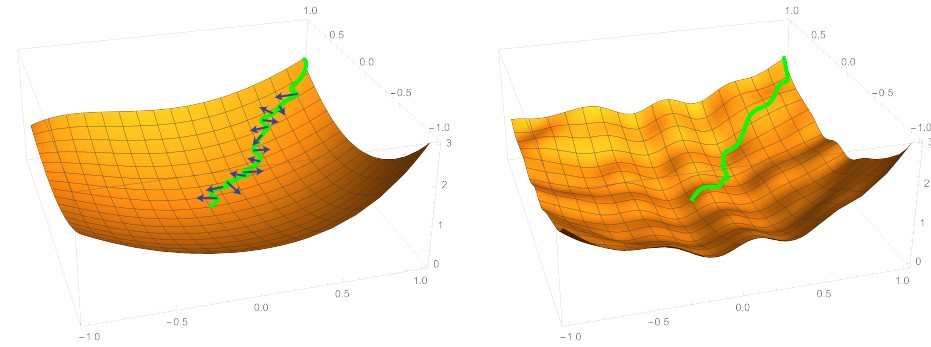

Figure 1: Left: Brownian motion — a particle driven by random forces within a static potential. Right: motion of a deterministic particle in a fluctuating potential.

The corresponding SGD update then becomes

$$\vec{w}_{n+1} = \vec{w}_n - \eta\vec{\nabla}\bar{L}(\vec{w}_n) - \eta\vec{\nabla}\delta L_n(\vec{w}_n).$$

Importantly, the deterministic component governed by $\bar{L}(\vec{w}_n)$ is exactly gradient descent (GD), while the additional term $-\eta\nabla\delta L_n(\vec{w}_n)$ captures the randomness of minibatch sampling.

**Master equation.** To move from individual update rules to a statistical description of the dynamics, we introduce the probability distribution $P_n(\vec{w})$ of the network parameters $\vec{w}$ at iteration $n$. This object characterizes the probability of finding the parameters in a given region of weight space after $n$ steps. Its evolution thus provides a complete description of the stochastic process defined by SGD.

A standard tool for such problems is the *master equation*, which encodes all possible transitions of the system between successive steps Risken (1989); Lucchi et al. (2022); Tan et al. (2023). For SGD, it takes the form

$$P_{n+1}(\vec{w}) = \int d^N v \, P_n(\vec{v}) \left\langle \delta^{(N)}\big(\vec{w} - \vec{v} + \eta\nabla\bar{L}(\vec{v}) + \eta\nabla\delta L(\vec{v})\big)\right\rangle_{\delta L}, \tag{2}$$

where $\delta^{(N)}(\vec{w})$ is the $N$-dimensional delta function and $N$ is the number of network parameters. The average $\langle\cdot\rangle_{\delta L}$ is taken over the minibatch-induced fluctuations of the loss $\delta L(\vec{w})$ (equation (1)).

A crucial aspect of this formulation is that fluctuations at different steps are *independent*, as each minibatch is sampled at random with replacement. This independence ensures that (2) provides an exact probabilistic description of SGD under this setting. In contrast, for SGD performed without replacement, correlations arise between consecutive updates, and a more complicated formulation of the master equation would be required.

**Fokker–Planck equation.** In statistical physics, the Fokker–Planck equation plays a central role in bridging microscopic stochastic dynamics with macroscopic statistical behavior (Risken, 1989; van Kampen, 2007). It describes how the probability distribution of a system evolves over time, and is widely used to study diffusion, relaxation to equilibrium, and fluctuation phenomena in systems ranging from Brownian particles to turbulent plasmas.

Adopting this perspective for optimization, we expand the master equation (2) in powers of the learning rate $\eta$ up to $\mathcal{O}(\eta^3)$, which yields a discrete Fokker–Planck equation for SGD (see Appendix A):

$$P_{n+1}(\vec{w}) \approx P_n(\vec{w}) + \eta \sum_{k=1}^{N} \nabla^k \big(P_n(\vec{w})\,\nabla^k\bar{L}(\vec{w})\big)$$
$$+ \frac{\eta^2}{2} \sum_{k,l=1}^{N} \nabla^k\nabla^l\Big(\big\langle\nabla^k L(\vec{w})\,\nabla^l L(\vec{w})\big\rangle_{\delta L} P_n(\vec{w})\Big) + \mathcal{O}(\eta^3). \tag{3}$$

Equation (3) provides a rigorous statistical description of how the probability distribution of network parameters evolves under SGD with replacement. Rather than tracking a single trajectory, it allows us to study the collective behavior across runs, to connect the structure of minibatch noise with loss landscape geometry, and to predict where SGD concentrates in parameter space.

A concrete set of insights that follow from equation (3) are:

- **Discrete nature.** The Fokker–Planck equation is derived in discrete form. Taking the strict continuous limit $\eta \to 0$ trivializes SGD to gradient descent, as the stochastic component vanishes with $\eta$.

- **Pitfalls of noise scaling.** Langevin-based approaches enforce $\sqrt{\eta}$ noise scaling to secure a continuous limit. This alters the diffusion coefficients, making them explicit functions of $\eta$ which results in neglecting key terms of the Fokker-Planck equation. Appendix A discusses this issue in detail.

- **Role of higher-order cumulants.** To leading order $\mathcal{O}(\eta^2)$, only the covariance (Gaussian part) of minibatch fluctuations contributes. Higher-order cumulants appear only as higher-order corrections $\mathcal{O}(\eta^3)$, unlike in Brownian models where they may play a significant role.

- **From Gaussian landscapes to non-Gaussian dynamics.** Even if minibatch fluctuations are Gaussian (e.g., in large-batch limits), they can still induce non-Gaussian dynamics of the parameters. Thus, observing heavy-tailed SGD trajectories does not necessarily imply heavy-tailed gradient noise.

In summary, equation (3) provides the main theoretical tool for the rest of this work. Since the only source of randomness in SGD originates from fluctuations of the loss function $L_n(\cdot)$, knowing the structure of the local curvature allows us to substitute this information directly into the Fokker–Planck equation. In the next section, we apply this approach by analyzing the dynamics near a loss critical point, where the landscape can be locally approximated by a quadratic form. This enables us to explicitly compute the mean and variance of SGD trajectories and to understand how curvature controls stability and generalization.

## 4 Motion Near Landscape Critical Point

**Loss function model near a critical point.** The Fokker–Planck equation (3) inherently depends on the correlators of the loss function's gradients. To make its analysis tractable, we approximate the loss landscape locally around a critical point of the averaged loss function. Let $\vec{v}$ denote such a point. Expanding to quadratic order gives

$$L(\vec{w}) = L_0 + \sum_{i=1}^{N} G_i(w_i - v_i) + \frac{1}{2} \sum_{i,j=1}^{N} H_{ij}(w_i - v_i)(w_j - v_j) + \ldots \quad (4)$$

where $G_i = \nabla^i L_n(\vec{v})$ are the stochastic gradient components and $H_{ij}$ is the stochastic Hessian at point $\vec{v}$. Averaging over minibatches yields

$$\langle L(\vec{w}) \rangle_{\delta L} = \langle L_0 \rangle_{\delta L} + \frac{1}{2} \sum_{i,j=1}^{N} \langle H_{ij} \rangle_{\delta L}(w_i - v_i)(w_j - v_j) + \ldots$$

The last equality holds since $\langle G_i \rangle_{\delta L} = 0$. Thus, $\vec{v}$ is a true critical point of the averaged loss, while for individual minibatches it may not coincide with an exact minimum. These fluctuations in gradients and curvature supply the correlators that enter the Fokker–Planck equation (3) and define the effective stochastic dynamics of SGD.

**Dispersion of weights trajectories.** We employ the Fokker-Planck equation (3) to study the statistical properties of neural network weight trajectories and to characterize the noise inherent in SGD. For this purpose, we define the following key quantities: the average parameters, measured from the critical point $\mu_i^n = \int d^N w P_n(\vec{w})(w_i - v_i)$ and their variance matrix

$$\Pi_{ij}^n = \int d^N w P_n(\vec{w}) w_i w_j - \mu_i^n \mu_j^n.$$

The variance matrix $\Pi_{ij}$ is a central object of our analysis, as it directly quantifies how individual trajectories deviate from the ensemble average. Using the quadratic loss model (4) introduced above, one can substitute it into the Fokker–Planck equation (3) and derive an analogous evolution equation for $\Pi_{ij}^{n+1}$. Using the recurrence relation for $\Pi_{ii}$ between steps $n$ and $n+1$, one can obtain a closed-form solution for $\Pi_{ii}$ (see equation (5)). Before doing so, however, an additional technical step is required. The explicit form of this equation is given in Appendix B.

**Mean Hessian eigenbasis.** The explicit Fokker–Planck–type equation for the variance matrix $\Pi_{ij}$ contains a large number of non-trivial cross terms. This makes the direct form analytically inconvenient. A significant simplification arises when the analysis is carried out in the eigenbasis of the mean Hessian. Formally, the eigenbasis of the mean Hessian is defined as $\langle H_{ij}\rangle_{\delta L} = \sum_{k=1}^{N} \lambda_k O_{ki} O_{kj}$, where $O_{ki}, O_{kj}$ are orthogonal matrices. Here $\lambda$ are eigenvalues of the mean Hessian $\langle H_{ij}\rangle_{\delta L}$. For this new rotated basis the variance matrix becomes

$$\tilde{H}_{ij} = \sum_{k,l=1}^{N} O_{ik} O_{jl} H_{kl}, \qquad \tilde{\Pi}_{ij}^n = \sum_{k,l=1}^{N} O_{ik} O_{jl} \Pi_{kl}^n.$$

Note that each $\tilde{H}_{ij}$ - minibatch Hessian rotated by basis of mean Hessian - is an almost diagonal matrix small non-diagonal elements corresponding to the SGD noise.

**Assumptions.** Received equation on $\tilde{\Pi}_{ij}$ are quite general. To make them tractable for analysis and solution, we introduce a set of reasonable assumptions (see Appendix B). The core assumptions of our model are:

1. Independence of fluctuations of all $\tilde{H}_{ij}$ components (as in Feng & Tu (2021a)).

2. Predominant diagonality of $\langle \tilde{H}_{ij}^2 \rangle_{\delta L}$.

3. Smallness of the learning rate $\eta \lambda^{max} < 1$. Here $\lambda^{max}$ is a maximum eigenvalue of the mean hessian $\langle H_{ij}\rangle_{\delta L}$.

These assumptions are basic and commonly used in analyses of SGD. The first two reflect the empirical structure of minibatch noise, while (3) is nothing more than the usual requirement that the step size is smaller than the inverse curvature scale $1/L$ of the loss landscape.

Utilizing the assumptions outlined above, we compute the variance matrix for the NN parameters in the vicinity of a critical point of the loss function (see the Appendix B for detailed calculations).

> **Key theoretical result of the paper.** In the basis of mean Hessian $\langle H_{ij}\rangle_{\delta L}$ the variance matrix $\tilde{\Pi}_{ij}^n$ is predominantly diagonal with the elements given by
>
> $$\tilde{\Pi}_{ii}^n = \eta\gamma \frac{1 - \left(1 - 2\eta\lambda_i + \eta^2 \langle \tilde{H}_{ii}^2 \rangle_{\delta L}\right)^n}{2\lambda_i - \eta \langle \tilde{H}_{ii}^2 \rangle_{\delta L}} (\lambda_i + \epsilon), \tag{5}$$
>
> where $\gamma$ - is some constant independent on learning rate, which may depends on batch size, and $\epsilon$ - is some very small regularizing constant which makes $\lambda_i + \epsilon$ positive for all directions, $n$ - the iteration step.

**Limiting cases.** Let us consider two limiting cases of the received expression. First, when the eigenvalue is very small $\lambda_i \sim 0$ we can show that $\tilde{\Pi}_{ii} \approx n\eta|\lambda_i|$. The variance along this direction very slowly grows linearly with time (iteration steps $n$). We classify such directions as diffusive or flat, corresponding to valleys in the loss landscape. Second, when the eigenvalue is sufficiently large $\lambda_i \gg 1$, the variance converges to a constant value $\tilde{\Pi}_{ii} \approx \frac{1}{2}\eta\gamma$, where $\gamma$ is some constant. Such directions correspond to rapidly changing features of the landscape and are termed rigid or sharp. Notably, for these rigid directions the inverse Einstein relation holds (Feng & Tu, 2021a). Other directions, however, are intermediate and lie between these two extremes, governed by the general solution given in equation (5) (see Appendix B for the full derivation).

In the following section, we confront our key theoretical result with empirical data from specific neural network models. This comparison serves to quantify the theory's predictive accuracy and to provide an intuitive explanation for the various aspects of the derived formula.

## 5 EXPERIMENTS

To evaluate our theoretical framework, we perform controlled experiments in both computer vision and natural language processing domains. The objective is to test directly whether the main predictions of the theory manifest in practice.

Specifically, we aim to demonstrate:

- the *inverse Einstein relation*: variance along directions of positive curvature grows initially and then saturates at a finite value;
- the absence of stationarity in nearly-flat directions, where trajectories keep diffusing indefinitely;
- the robustness of these effects across architectures and data modalities.

Our experimental protocol is the same in both CV and NLP domains (for the full description of the experiments see Appendix D):

1. Train a model using SGD until the loss is well minimized.
2. From this trained point, continue optimization with GD to converge fully to a local minimum.
3. Along the SGD trajectories, compute Hessians of the loss and project the dynamics into the Hessian eigenbasis.
4. Compare the measured parameter variance with theoretical predictions and analyze how different eigendirections behave.

This setup allows us to disentangle the dynamical behavior of SGD across different curvature regimes and to assess how closely the observed trajectories follow our theoretical predictions.

To directly evaluate our theoretical predictions, we perform experiments on small and medium-sized models, where the Hessian and minibatch covariance can be computed exactly. This restriction is purely computational: full Hessian analysis is infeasible for modern large-scale architectures.

Importantly, the curvature regime underlying our analysis is not tied to small models. Prior empirical evidence, in particular (Feng & Tu, 2021a), shows that the qualitative Hessian structure remains stable as network size increases. Specifically, they argue that increasing the network width does not increase the number of sharp directions, and that the additional degrees of freedom introduced by wider architectures remain flat and high-flatness. Their results, verified on convolutional networks trained on MNIST and CIFAR-10, support that the sharp–flat separation persists in larger and modern architectures.

### 5.1 COMPUTER VISION

We empirically evaluate the theoretical predictions on the MNIST (LeCun et al., 2010) dataset using fully connected networks (MLPs) (Rosenblatt, 1958) with two layers and learning rate $\eta = 0.05$. Starting from a stable minimum of the loss landscape, we compute 1000 Hessians along the SGD trajectory (in total across 386 parameters; see Appendix D for details). Averaging over minibatches yields the mean Hessian $\langle H_{ij} \rangle_{\delta L}$, whose eigenvalue spectrum is shown in Fig. 2 (left).

The spectrum reveals that most eigenvalues are clustered near zero or slightly negative, with only a few directions exhibiting significant positive curvature ($\lambda^{\max} \approx 6$). Hence the landscape is dominated by wide, flat valleys, interrupted by a small set of sharper directions.

To connect this structure with SGD dynamics, we generated 1000 independent trajectories from the same minimum. Rotating these trajectories into the eigenbasis of the mean Hessian, we computed the variance matrix, shown in Fig. 2 (right). The variance is nearly diagonal, which implies that Hessian eigenvectors align with statistically independent directions of SGD noise. Thus, eigenvalues of the

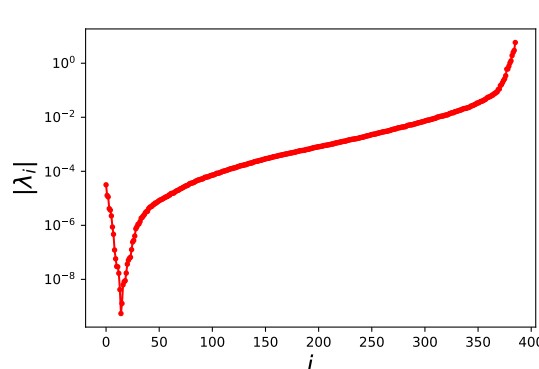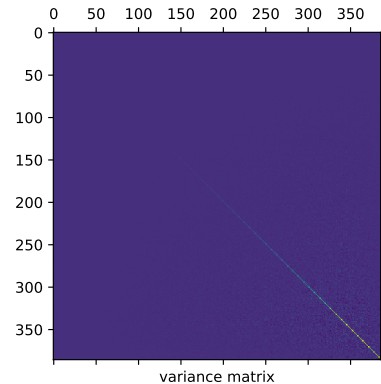

Figure 2: Left: eigenvalue spectrum of the mean Hessian. Right: variance matrix of SGD trajectories in the mean-Hessian eigenbasis.

mean Hessian provide a natural coordinate system for disentangling different modes of parameter evolution.

Additional illustrations of representative trajectories in flat versus curved directions are provided in Appendix C, where we explicitly confirm that flat eigendirections exhibit diffusive behavior, while positively curved modes remain confined. These results directly validate our theoretical framework and demonstrate that the Hessian spectrum partitions parameter space into qualitatively distinct regimes of SGD behavior.

Followed by this experiment, and as further illustrated with example trajectories in Appendix C, we identify two distinct types of directions in the parameter space:

**Diffusive directions** ($\lambda_i \approx 0$). In these eigendirections, the spread of parameters keeps increasing throughout the experiment, never settling down to a fixed range.

**Rigid directions (large $\lambda_i > 0$).** These directions have eigenvalues large enough that the parameter spread quickly reaches a stable range within the observed training time.

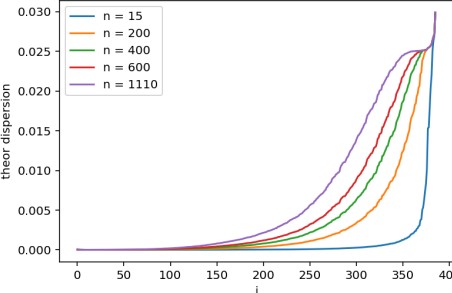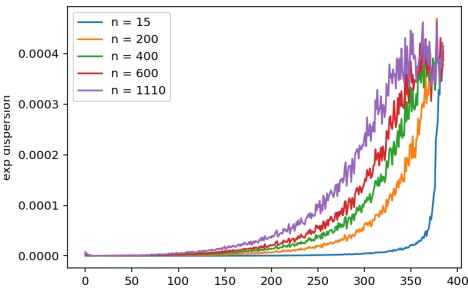

Figure 3: Left: theoretical prediction for the diagonal elements of the variance matrix from equation (5). Right: experimental measurement of the same quantity. Note that the theory is unable to predict the value of the overall multiplicative constant $\gamma$; therefore, the magnitudes of the theoretical and experimental curves do not coincide..

Figure 3 compares our theoretical prediction for the diagonal elements of the variance matrix (equation (5)) with empirical measurements. The result relies solely on Hessian information and correctly captures the general pattern and behavior of the parameter spread, with a numerical factor $\gamma$ that does not depend on the learning rate $\eta$ but can vary with batch size and network architecture.

This comparison reveals two important aspects. First, the system as a whole does not reach a truly stable state: directions with small $\lambda_i$ (diffusive modes) continue to spread indefinitely, while rigid modes stabilize at a finite spread of approximately $\frac{1}{2}\gamma\eta$ (see Appendix B). Second, different modes settle down at different rates: less rigid directions take longer to reach a stable spread. Because the Hessian eigenvalues form a continuous spectrum, achieving a state where all directions are fully stable would require an infinitely long training time.

Consequently, SGD dynamics do not reach a true stable state in practice. Since the network parameters $\vec{w}$ are a combination of both spreading and stable components, their overall spread consistently grows with training time. This means that any single SGD training run will increasingly diverge from the average behavior of many runs and will no longer be representative of the ensemble.

Finally, one might hope that this spreading behavior could help explore flat valleys in search of better minima. However, our calculations show that the rate of spread along a direction depends on its corresponding Hessian eigenvalue $\lambda_i$. In flat valleys, where $\lambda_i$ is very small, the effective spreading is also very slow. Instead of freely exploring these flat regions, the dynamics are dominated by movement along the stiffer directions, which can lead the SGD trajectory further away from the initially found minimum.

## 5.2 NATURAL LANGUAGE PROCESSING

To check that our results are not limited to computer vision, we repeat the analysis on a text modeling task. We train a simplified GPT (Radford & Narasimhan, 2018) model on the Shakespeare corpus (Karpathy, 2015). Here the task is set up as symbolic encoding: each word (or character) is shifted by one position in the alphabet, similar to the classical Caesar cipher (Wikipedia contributors). This makes the problem simple enough to allow a full Hessian analysis, while still keeping the sequential structure of language. For the language model, we repeated the same set of experiments as conducted

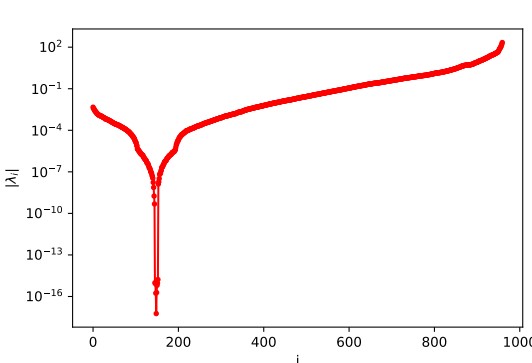
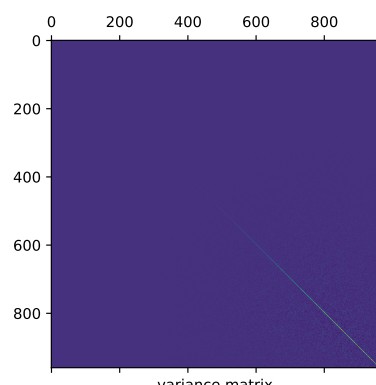

Figure 4: Left: eigenvalue spectrum of the mean Hessian. Right: variance matrix of parameter trajectories in the mean-Hessian eigenbasis.

for the MLP, with learning rate $\eta = 0.001$. The spectral structure differs in scale — the maximum eigenvalue is substantially larger, $\lambda^{\mathrm{max}} \approx 217$ (Fig. 4, left). Nevertheless, as in the vision case, most eigenvalues are near zero, indicating that the loss landscape is again dominated by wide flat valleys.

The variance matrix (Fig. 4, right) remains close to diagonal when expressed in the Hessian eigenbasis. This shows that parameter fluctuations decouple along eigendirections, and the distinction between diffusive and rigid modes derived in the previous section applies equally well here. Example trajectories confirming these behaviors are provided in Appendix C.

The comparison in Fig. 5 confirms that the theoretical framework remains valid in the language setting: rigid modes stabilize at a finite variance, while flat directions display ongoing spread. Thus, the separation of SGD dynamics into diffusive and rigid directions is a general property, not limited to a specific domain or architecture.

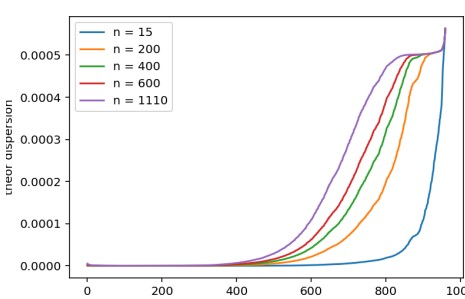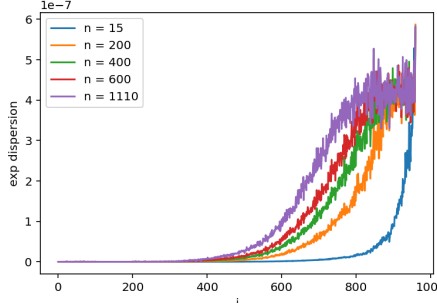

Figure 5: Left: theoretical prediction for the diagonal elements of the variance matrix from equation (5). Right: experimental measurement of the same quantity. Note that the theory is unable to predict the value of the overall multiplicative constant $\gamma$; therefore, the magnitudes of the theoretical and experimental curves do not coincide.

## 6 CONCLUSION

In this work, we revisited the stochastic dynamics of SGD by deriving its discrete Fokker–Planck equation directly from the update rule, without relying on continuous-time approximations. This formulation clarifies the origin and role of minibatch-induced fluctuations and highlights the correction terms that are typically lost in the Langevin-based picture.

By analyzing the dynamics near a critical point of the loss, we obtained an explicit characterization of variance evolution along Hessian eigendirections. This analysis reveals a clear separation between sharply curved directions, which admit a stationary variance, and nearly-flat directions, where no stationary state emerges and the dynamics exhibit persistent drift or diffusion.

We validated these theoretical predictions on controlled experimental settings where both the Hessian spectrum and the covariance structure of the stochastic gradient can be computed exactly. In particular, Figures 3 and 5 show that the variance dynamics along individual Hessian eigendirections closely follow the three regimes predicted by our theory: rapid saturation in high-curvature directions, slow growth near flat directions, and effective diffusion in valley-like regions.

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

# A    SGD Fokker-Planck equation and Langevin-like approach

This section is devoted the derivation of the Fokker-Planck recursion relation. Our starting point is the master equation, which relates probability distribution function at two subsequent iterations (equation(2)):

$$P_{n+1}(\vec{w}) = \int d^N v \, P_n(\vec{v}) \left\langle \delta^{(N)}\big(\vec{w} - \vec{v} + \eta \nabla L(\vec{v})\big)\right\rangle_{\delta L}. \tag{6}$$

Let us employ the representation of the delta function in terms of its Fourier transform

$$\delta^{(N)}(\vec{a}) = \int \frac{d^N p}{(2\pi)^N} e^{i \sum_{i=1}^N p_i a^i}. \tag{7}$$

This leads to the relation

$$P_{n+1}(\vec{w}) = \int \frac{d^N v d^N p}{(2\pi)^N} \, P_n(\vec{v}) \, e^{i \sum_{i=1}^N p_i (w^i - v^i)} \left\langle e^{i\eta \sum_{i=1}^N p_i \nabla^i L(\vec{v})}\right\rangle_{\delta L}. \tag{8}$$

Now it is possible to expand the above expression in $\eta$

$$P_{n+1}(\vec{w}) = \sum_{m=0}^{\infty} \frac{(i\eta)^m}{m!} \sum_{i_1,..,i_m=1}^N \int \frac{d^N v d^N p}{(2\pi)^N} \, P_n(\vec{v}) \, e^{i \sum_{i=1}^N p_i (w^i - v^i)} p_{i_1}...p_{i_m}$$

$$\times \left\langle \nabla^{i_1} L(\vec{v})...\nabla^{i_m} L(\vec{v})\right\rangle_{\delta L}, \tag{9}$$

and perform all the integrals over $\vec{p}$ with the help of expression:

$$\int \frac{d^N p}{(2\pi)^N} \, e^{i \sum_{i=1}^N p_i (w^i - v^i)} p_{i_1}...p_{i_m} = (-i)^m \nabla^{i_1}...\nabla^{i_m} \delta^{(N)}(\vec{w} - \vec{v}). \tag{10}$$

Then the resulting recursion relation is

$$P_{n+1}(\vec{w}) = \sum_{m=0}^{\infty} \frac{\eta^m}{m!} \sum_{i_1,..,i_m=1}^N \nabla^{i_1}...\nabla^{i_m} \Big( \left\langle \nabla^{i_1} L(\vec{w})...\nabla^{i_m} L(\vec{w})\right\rangle_{\delta L} P_n(\vec{w}) \Big). \tag{11}$$

To the second order it is represented in the main text.

Now let us compare our approach with the naive treatment of SGD in terms of Langevin equation. Let us recall again the one SGD step

$$w_{n+1}^i = w_n^i - \eta \nabla^i L(\vec{w}_n). \tag{12}$$

The commonly used approach consists of separation of minibatch loss into average and fluctuations and model the SGD step by Langevin-like equation

$$w_{n+1}^i = w_n^i - \eta \nabla^i \bar{L}(\vec{w}_n) - \sqrt{\eta} \sum_{j=1}^N C^{ij}(\vec{w}_n) \xi_n^j, \tag{13}$$

where $\vec{\xi}_n$ - is an uncorrelated Gaussian process with zero mean and unit variance $\langle \xi_n^i \xi_m^j \rangle_\xi = \delta_{ij} \delta_{mn}$. Matrix $C^{ij}(\vec{w})$ is restored by the comparison of noise correlators with the loss function fluctuations.

$$\sum_{k=1}^N C^{ik}(\vec{w}^n) C^{kj}(\vec{w}_n) = \eta \langle \nabla^i L(\vec{w}) \nabla^j L(\vec{w}) \rangle_{\delta L} - \eta \nabla^i \bar{L}(\vec{w}) \nabla^j \bar{L}(\vec{w}) \equiv W^{ij}(\vec{w}). \tag{14}$$

After that we may proceed in the same way as above and use master equation

$$P_{n+1}(\vec{w}) = \int d^N v \left\langle \delta^{(N)}\Big( w^i - v^i + \eta \nabla^i \bar{L}(v) + \sqrt{\eta} \sum_{j=1}^N C^{ij}(v) \xi_n^j \Big)\right\rangle_\xi P_n(\vec{v}), \tag{15}$$

along with delta function representation

$$P_{n+1}(\vec{w}) = \int \frac{d^N v d^N p}{(2\pi)^N} \, e^{i \sum_{i=1}^N p_i (w^i - v^i + \eta \nabla^i \bar{L}(v))} \left\langle e^{i\sqrt{\eta} \sum_{i,j=1}^N p_i C^{ij}(v) \xi_n^j}\right\rangle_\xi P_n(\vec{v}). \tag{16}$$

After the averaging over $\vec{\xi}$ the one step relation becomes

$$P_{n+1}(\vec{w}) = \int \frac{d^N v d^N p}{(2\pi)^N} \, e^{i\sum_{i=1}^N p_i(w^i - v^i) + i\eta \sum_{i=1}^N p_i \nabla^i \bar{L}(v) - \eta \sum_{i,j=1}^N W^{ij}(v)p_i p_j} \, P_n(\vec{v}). \quad (17)$$

For small $\eta$ one has the following Fokker-Planck recursion

$$P_{n+1}(\vec{w}) = P_n(\vec{w}) + \eta \sum_{k=1}^N \nabla^k \left( P_n(\vec{w}) \nabla^k \bar{L}(\vec{w}) \right) + \frac{1}{2}\eta \sum_{k,l=1}^N \nabla^k \nabla^l \left( W^{kl}(\vec{w}) P_n(\vec{w}) \right) + .... \quad (18)$$

Let us compare it with the equation (11). This equation looks similar but one term $\sim \nabla^k \nabla^l \left( \nabla^k \bar{L}(\vec{w}) \nabla^l \bar{L}(\vec{w}) P_n(\vec{w}) \right)$ is missed. This is due to the fact that $W^{kl} \sim \eta$ and the last term in equation (18) is of the $\eta^2$ order, not $\eta$. It means that for consistency one should add all other terms of the same order $\eta^2$. In the present case there is only one term, which we are mention above. But for non-gaussian noise and/or higher order correction in the Langevin-like approach there is a mix of different kind of contributions in different orders which should be kept. The developed by us approach is much more clear and straightforward.

## B  MOTION NEAR THE LANDSCAPE CRITICAL POINT: COMPLETE CALCULATIONS

Our starting point is the Fokker-Planck equation, which is

$$P_{n+1}(\vec{w}) = P_n(\vec{w}) + \eta \sum_{k=1}^N \nabla^k \left( P_n(\vec{w}) \nabla^k \bar{L}(\vec{w}) \right)$$

$$+ \frac{1}{2}\eta^2 \sum_{k,l=1}^N \nabla^k \nabla^l \left( \langle \nabla^k L(\vec{w}) \nabla^l L(\vec{w}) \rangle_{\delta L} P_n(\vec{w}) \right) + .... \quad (19)$$

In order to understand the behavior in the exploration phase let us approximate loss near some critical point $\vec{w}_c = \vec{v}$ by quadratic function

$$L(\vec{w}) = L_0 + \sum_{i=1}^N G_i(w^i - v^i) + \frac{1}{2} \sum_{i,j=1}^N H_{ij}(w^i - v^i)(w^j - v^j) + .... \quad (20)$$

Here $L_0$, $G_i$ $H_{ij}$ are the parameters of the loss function which are fluctuate from one minibatch to another. The average loss is

$$\langle L(\vec{w}) \rangle_{\delta L} = \langle L_0 \rangle_{\delta L} + \frac{1}{2} \sum_{i,j=1}^N \langle H_{ij} \rangle_{\delta L}(w^i - v^i)(w^j - v^j) + .... \quad (21)$$

so $\langle G_i \rangle_{\delta L} = 0$. For this case one can derive the closed set of equations for the average parameters (measured from the critical point)

$$\mu_i^n = \int d^N w P_n(\vec{w})(w^i - v^i) \quad (22)$$

and their variance

$$\Pi_{ij}^n = \int d^N w P_n(\vec{w}) w^i w^j - \mu_n^i \mu_n^j. \quad (23)$$

It has the form

$$\mu_{n+1}^i = \mu_n^i - \eta \sum_{j=1}^N \langle H_{ij} \rangle_{\delta L} \mu_n^j, \quad (24)$$

$$\Pi_{n+1}^{ij} = \Pi_n^{ij} - \eta \sum_{k=1}^N \left( \Pi_n^{ik} \langle H_{kj} \rangle_{\delta L} + \langle H_{ik} \rangle_{\delta L} \Pi_n^{kj} \right) + \eta^2 \sum_{k,l=1}^N \langle H_{ik} H_{jl} \rangle_{\delta L} \Pi_n^{kl} + \eta^2 \Lambda_n^{ij}, \quad (25)$$

where

$$\Lambda_n^{ij} = \left\langle \left(G_i + \sum_{k=1}^{N} H_{ik}\mu_n^k\right)\left(G_j + \sum_{l=1}^{N} H_{jl}\mu_n^l\right)\right\rangle_{\delta L} - \sum_{k,l=1}^{N} \langle H_{ik}\rangle_{\delta L}\langle H_{jl}\rangle_{\delta L}\mu_n^k\mu_n^l. \quad (26)$$

Before analyzing these equations let us introduce the eigenbasis of $\langle H_{ij}\rangle_{\delta L} = \sum_{k=1}^{N}\lambda_k O_{ki}O_{kj}$, where $O_{ij}$ is orthogonal, and rotate the equations to it. Here $\lambda_i$ are the eigenvalues of $\langle H_{ij}\rangle_{\delta L}$. In rotated basis

$$\tilde{G}_i = \sum_{k=1}^{N} O_{ik}G_k, \quad \tilde{H}_{ij} = \sum_{k,l=1}^{N} O_{ik}O_{jl}H_{kl}, \quad \tilde{\mu}_n^i = \sum_{k=1}^{N} O_{ik}\mu_n^k, \quad \tilde{\Pi}_n^{ij} = \sum_{k,l=1}^{N} O_{ik}O_{jl}\Pi_n^{kl}. \tag{27}$$

and the equations (24), (25) have the form

$$\tilde{\mu}_{n+1}^i = \tilde{\mu}_n^i - \eta\lambda_i\tilde{\mu}_n^i, \tag{28}$$

$$\tilde{\Pi}_{n+1}^{ij} = \tilde{\Pi}_n^{ij} - \eta(\lambda_i + \lambda_j)\tilde{\Pi}_n^{ij} + \eta^2\sum_{k,l=1}^{N}\langle\tilde{H}_{ik}\tilde{H}_{jl}\rangle_{\delta L}\tilde{\Pi}_n^{kl} + \eta^2\tilde{\Lambda}_n^{ij}, \tag{29}$$

Detailed analysis of these equation will be published in the another paper, but here we make simplifying assumptions, which are rather natural for NN under consideration. Namely we assume that fluctuations of all components of $\tilde{G}_i$ and $\tilde{H}_{ij}$ are independent. In other words

$$\langle\tilde{G}_i\tilde{G}_j\rangle_{\delta L} = d_i\delta_{ij}, \qquad \langle\tilde{H}_{ij}\rangle_{\delta L} = \lambda_i\delta_{ij}, \tag{30}$$

$$\langle\tilde{H}_{ik}\tilde{H}_{jl}\rangle_{\delta L} = \lambda_i\lambda_j\delta_{ik}\delta_{jl} + \Gamma_{ik}\big(\delta_{ij}\delta_{kl} + \delta_{il}\delta_{jk} - \delta_{ij}\delta_{ik}\delta_{jl}\big). \tag{31}$$

Here $\Gamma_{ij}$ is nothing but the variance of fluctuations of $i,j$-th element of matrix $\tilde{H}_{ij}$. At first let us solve equation for the average values of parameters. Its solution is $\tilde{\mu}_n^i = (1 - \eta\lambda_i)^n\tilde{\mu}_0^i$, where $\mu_0^i$ is the initial point. From this equation one can obtain

$$\tilde{\Lambda}_n^{ij} = (1 - \eta\lambda_i)^n(1 - \eta\lambda_j)^n(1 - \delta_{ij})\Gamma_{ij}\tilde{\mu}_0^i\tilde{\mu}_0^j + \delta_{ij}\sum_{k=1}^{N}\Gamma_{ik}(1 - \eta\lambda_k)^{2n}(\tilde{\mu}_0^k)^2 + d_i\delta_{ij}. \quad (32)$$

Now one can solve the equation for the variance. At first consider the non-diagonal elements $i \neq j$. In this case

$$\tilde{\Pi}_{n+1}^{ij} = \big((1 - \eta\lambda_i)(1 - \eta\lambda_i) + \eta^2\Gamma_{ij}\big)\tilde{\Pi}_n^{ij} + \eta^2(1 - \eta\lambda_i)^n(1 - \eta\lambda_j)^n\Gamma_{ij}\mu_0^i\tilde{\mu}_0^j, \quad (33)$$

which can be solved as

$$\tilde{\Pi}_n^{ij} = \Big(\big((1 - \eta\lambda_i)(1 - \eta\lambda_i) + \eta^2\Gamma_{ij}\big)^n - (1 - \eta\lambda_i)^n(1 - \eta\lambda_j)^n\Big)\tilde{\mu}_0^i\tilde{\mu}_0^j. \tag{34}$$

For the diagonal elements one has the equation

$$\tilde{\Pi}_{n+1}^{ii} = (1 - \eta\lambda_i)^2\tilde{\Pi}_n^{ii} + \eta^2\sum_{k=1}^{N}\Gamma_{ik}\Big(\tilde{\Pi}_n^{kk} + (1 - \eta\lambda_k)^{2n}(\tilde{\mu}_0^k)^2\Big) + \eta^2 d_i. \tag{35}$$

This equation is more complicated, however it can be solved in the matrix form. Let us introduce the diagonal matrix $\hat{L}$ with elements $L_{ij} = (1 - \eta\lambda_i)^2\delta_{ij}$ and the matrix $\hat{\Gamma}$ with elements $\Gamma_{ij}$. Then the solution is

$$\tilde{\Pi}_n^{ii} = \sum_{k=1}^{N}\big[(\hat{L} + \eta^2\hat{\Gamma})^n - \hat{L}^n\big]_{ik}(\tilde{\mu}_0^k)^2 + \eta^2\sum_{k=1}^{N}\left[\frac{1 - (\hat{L} + \eta^2\hat{\Gamma})^n}{1 - \hat{L} - \eta^2\hat{\Gamma}}\right]_{ik}d_k. \tag{36}$$

Solutions (34),(36) have two contributions to the variance matrix $\tilde{\Pi}_n^{ij}$. The first one is related to the initial conditions, whereas the second one is diagonal (in the basis chosen). Now let us analyze this general result in context of numerical experiments made by us.

Typically in our experiments the fluctuation matrix $\eta^2\Gamma$ are rather small (but not negligible) and the matrix $\Gamma_{ij}$ itself are nearly diagonal. We observe that one can safely neglect the non-diagonal

elements of $\Gamma_{ij}$. Also one can check, that the term containing initial condition are unimportant. Then the variance matrix $\tilde{\Pi}_{ij}^n$ is diagonal in the basis composed of $\langle H_{ij}\rangle_{\delta L}$ eigenvectors. This is supported by our experiments (see the main text). In other word in this basis all the directions evolve independently and their behavior depends on the eigenvalues $\lambda_i$. For near-zero eigenvalues the $(1-\eta\lambda_i)^n$ quantity might be still near 1 up to step $n$, so one can expand the variance in $\eta\lambda_i$. After the simple algebra we obtain $\Pi_n^{ii} \approx \eta^2 n d_k$ if $\eta n |\lambda_i| \ll 1$. These directions are non-stationary and the system randomly moves in this directions like Brownian particle. Although the effective diffusion coefficient $\eta d_k$ occurs to be very small, the large number of these nearly-flat directions leads to a noticeable divergence of individual trajectories. In the directions with large eigenvalues $\eta n \lambda_i \gg 1$ the system reaches the stationary state given by

$$\Pi_n^{ii} \approx \frac{\eta d_i}{2\lambda_i - \eta\lambda_i^2 - \eta\Gamma_{ii}}. \tag{37}$$

The behavior in the other (intermediate) directions is in between these two limiting cases.

There is an extremely interesting phenomena, which we see in our numerical experiments. We observe that for directions with positive eigenvalue there is a proportionality $d_i \sim \lambda_i$. In other words $\langle G_i G_j\rangle_{\delta L} \approx \gamma(\langle H_{ij}\rangle_{\delta L} + \epsilon_{ij})$, where the matrix $\epsilon_{ij}$ is small and necessary only for the regularization of negative eigenvalues. $\gamma$ is a some constant, which is independent form the learning rate. We propose some mechanism of this phenomena later, but now we substitute this relation into the result for large $\lambda_i$ directions

$$\Pi_{ii}^n \approx \frac{\eta\gamma\lambda_i}{2\lambda_i - \eta\lambda_i^2 - \eta\Gamma_{ii}} \approx \frac{\eta\gamma}{2}. \tag{38}$$

We obtain the following picture. Near the critical point, there are many directions with small eigenvalues in which the system moves chaotically, resulting in divergence of trajectories. Whereas in the the rigid direction there is something like the evolution to the stationary state, but the width of this stationary distribution is almost the same for all (rigid) directions, i.e. independent from $\lambda_i$. For the intermediate directions the variance is depended on the iteration number $n$. Under the assumptions, stated above it is necessary to consider only diagonal elements of matrix $\hat{L} + \eta^2\hat{\Gamma}$, which are

$$(1-\eta\lambda_i)^2 + \eta^2\Gamma_{ii} = 1 - 2\eta\lambda_i + \eta^2\langle\tilde{H}_{ii}^2\rangle_{\delta L}. \tag{39}$$

The last equality follows from the definition of $\Gamma_{ij}$. Putting all together one can reproduce the main result of our paper from the equation (36)

$$\tilde{\Pi}_{ii}^n = \eta\gamma\frac{1 - \left(1 - 2\eta\lambda_i + \eta^2\langle\tilde{H}_{ii}^2\rangle_{\delta L}\right)^n}{2\lambda_i - \eta\langle\tilde{H}_{ii}^2\rangle_{\delta L}}(\lambda_i + \epsilon) \tag{40}$$

This result naturally explains the inverse Einstein relation discussed earlier in the literature. It is just the non-stationary effect. The effective fluctuations in the direction of small $\lambda_i$ are small and system are not evolve in this directions resulting in the small distribution width at the finite step.

At the end of this appendix let us make few comments about the origin of $\langle G_i G_j\rangle_{\delta L} \sim \langle H_{ij}\rangle_{\delta L}$ relation. One of the possible mechanism is the following. Consider the minibatch loss near the critical point in the form

$$L(\vec{w}) = \frac{1}{N_B}\sum_{k=1}^{N_B}\Phi\left(\sum_{i=1}^{N}b_i^{(k)}(w_i - v_i)\right), \tag{41}$$

where $N_B$ is a minibatch size and $\vec{b}^{(k)}$ is a random vector which is attributed to the given sample. $\Phi(x)$ - is some function. Expanding it around the critical point one obtain

$$L(\vec{w}) \approx \Phi(0) + \frac{1}{N_B}\Phi'(0)\sum_{k=1}^{N_B}\sum_{i=1}^{N}b_i^{(k)}(w_i - v_i)$$

$$+ \frac{1}{2N_B}\Phi''(0)\sum_{k=1}^{N_B}\sum_{i,j=1}^{N}b_i^{(k)}b_j^{(k)}(w_i - v_i)(w_j - v_j) + ... \tag{42}$$

From this one can identify

$$G_i = \frac{1}{N_B}\Phi'(0)\sum_{k=1}^{N_B}b_i^{(k)}, \qquad H_{ij} = \frac{1}{N_B}\Phi''(0)\sum_{k=1}^{N_B}b_i^{(k)}b_j^{(k)}, \tag{43}$$

so the relation

$$\langle G_i G_j \rangle_b = \frac{1}{N_B} \left( \Phi'(0) \right)^2 \langle b_i b_j \rangle_b = \frac{\left( \Phi'(0) \right)^2}{N_B \Phi''(0)} \langle H_{ij} \rangle_b \tag{44}$$

holds. Definitely this is only an example, but it mimics the basic idea. The fluctuations of both the gradient and the Hessian are originated from the same source, so they may share the same information about the randomness of the system and be proportional to the same matrix.

## MAIN RESULT

The main result of the paper

$$\tilde{\Pi}_{ii}^n = \eta \gamma \frac{1 - \left( 1 - 2\eta\lambda_i + \eta^2 \langle \tilde{H}_{ii}^2 \rangle_{\delta L} \right)^n}{2\lambda_i - \eta \langle \tilde{H}_{ii}^2 \rangle_{\delta L}} (\lambda_i + \epsilon), \tag{45}$$

where $\gamma$ - is some constant independent on learning rate, which may depends on the batch size, and $\epsilon$ - is some very small regularizing constant which makes $\lambda_i + \epsilon$ positive for all directions.

# C  SGD TRAJECTORIES IN DIFFERENT EIGENDIRECTIONS

To illustrate the distinct behaviors predicted by our theoretical framework, we examine individual SGD trajectories projected onto representative eigendirections of the mean Hessian. Results are shown both for the MLP experiment (vision) and the Shakespeare model (language).

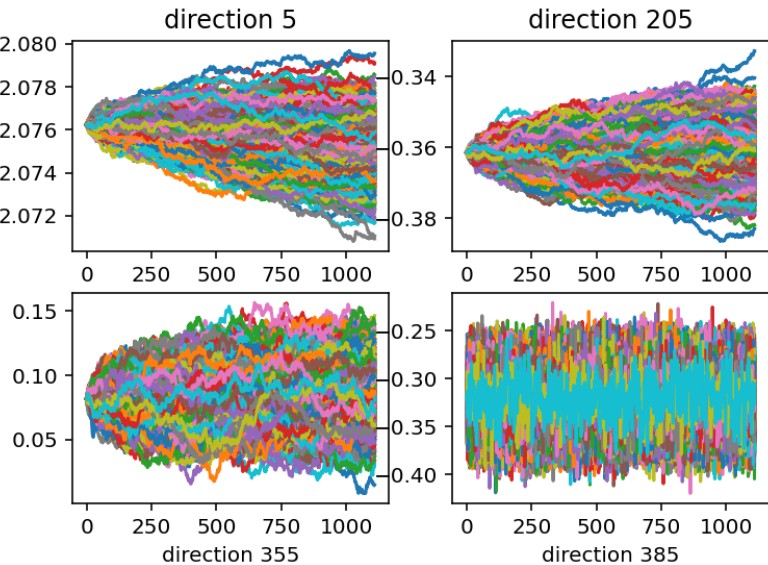

Figure 6: MLP experiment: trajectories in distinct eigendirections of the mean Hessian. **Top:** flat directions ($\lambda \approx 0$) show diffusive motion with variance growing in time. **Bottom:** curved directions ($\lambda > 0$) remain confined, with variance saturating at a finite value.

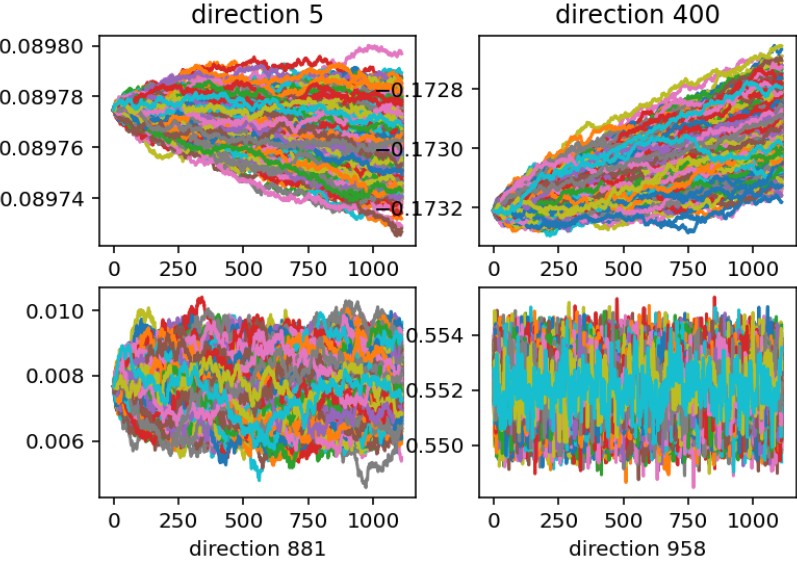

Figure 7: Shakespeare model: trajectories in distinct eigendirections of the mean Hessian. **Top:** flat directions exhibit diffusive drift. **Bottom:** curved directions show confined dynamics, consistent with the MLP case.

**Diffusive directions (top panels).** For both vision and language tasks, in eigendirections corresponding to near-zero or slightly negative eigenvalues, SGD trajectories display unconfined motion. Variance grows steadily in time, consistent with diffusion along valley floors and in agreement with the Fokker–Planck analysis.

**Rigid directions (bottom panels).** In directions with large positive curvature, trajectories remain localized around the minimum. The variance quickly saturates to a finite value controlled by the balance between curvature and minibatch noise. Notably, sharper directions exhibit larger fluctuations, directly confirming the inverse Einstein relation $\mathrm{Var}[\Delta w] \propto \lambda^{-1}$.

**Cross-task consistency.** The similarity of behaviors between MLP and Shakespeare strongly supports the universality of our framework: the decomposition of SGD dynamics into diffusive and rigid modes is not tied to the specific architecture or data domain, but instead appears to be a general property of optimization in high-dimensional loss landscapes.

# D EXPERIMENT SETUPS

Our code is available at `https://anonymous.4open.science/r/SGDiffusion`.

## D.1 EXPERIMENT SETUP: MNIST

We study the behavior of SGD and GD on the MNIST (LeCun et al., 2010) dataset using a multilayer perceptron (MLP) (Rosenblatt, 1958) with 386 parameters. The training protocol consists of two long phases: (i) SGD training from random initialization for $10^4$ epochs, followed by (ii) GD training for another $10^4$ epochs starting from the final SGD checkpoint. This produces a stable reference point from which further analyses are conducted.

To analyze optimization dynamics, we consider three complementary experiments:

1. **SGD trajectories:** multiple independent runs from the same starting point with different learning rates and random seeds, collecting 1000 trajectories, each of length 1111 mini-batch iterations.

2. **GD trajectories:** deterministic full-gradient runs with different learning rates, each consisting of 1111 steps.

3. **Hessian analysis:** additional SGD runs of 1000 iterations, during which Hessians are computed along the trajectory.

Representative configuration: MLP on MNIST with batch size 64 and learning rate 0.1, trained on a reduced dataset of 6400 samples (the first 6400 elements of the standard MNIST training set). Training uses *sampling with replacement*, meaning that after each mini-batch, the drawn samples are returned to the dataset pool and may be selected again in subsequent iterations. This emphasizes the role of repeated sampling in shaping the stochasticity of SGD trajectories compared to the standard epoch-wise sampling without replacement. We deliberately use such small datasets to rapidly overfit compact models and to make the subsequent analysis of their local minima tractable.

**MLP architecture.** We use a compact multilayer perceptron with optional input downsampling, one hidden layer, and a final linear classifier. In our experiments, we set the hidden dimension to 8, the number of hidden layers to 1, and apply downsampling of the $28 \times 28$ input image to $6 \times 6$ (flattened to 36 features). Thus, the model consists of:

- input layer: $36 \to 8$,
- hidden nonlinearity (SiLU),
- output layer: $8 \to 10$.

This results in a total of 386 trainable parameters.

## D.2 EXPERIMENT SETUP: SHAKESPEARE

We study the behavior of SGD and GD in the setting of character-level language modeling using the Tiny Shakespeare (Karpathy, 2015) dataset and a lightweight NanoGPT (Radford & Narasimhan, 2018) architecture with 960 parameters. The training protocol again consists of two long phases: (i) SGD training from random initialization for $10^4$ epochs with learning rate 0.01, followed by (ii) GD training for another $10^4$ epochs starting from the final SGD checkpoint. This produces a stable reference point from which further analyses are conducted. We use a reduced dataset of 6000 characters, which allows the small model to rapidly overfit and makes the subsequent analysis of its local minima tractable.

We then perform the same three complementary experiments:

1. **SGD trajectories:** 1000 runs of 1111 mini-batch iterations with varying learning rates and seeds,

2. **GD trajectories:** 1111 full-gradient steps with different learning rates,

3. **Hessian analysis:** 1000-iteration SGD runs with Hessians computed along the trajectory.

**Shakespeare dataset.** We construct a character-level dataset from the Tiny Shakespeare corpus. The text is truncated to 6000 characters and mapped to a restricted alphabet of 25 symbols ("a"–"y" plus space), with all other characters replaced by space. Each sequence is tokenized at the character level and segmented into overlapping blocks of length 16, where the model predicts the next character given the previous context. This yields approximately 5984 sequences, split into 80% training and 20% validation (4787 train and 1197 validation samples).

For training, we use a dataloader with *sampling with replacement*, where after each mini-batch the drawn samples are returned to the dataset pool.

**NanoGPT architecture.** For the Shakespeare experiments, we employ a minimal transformer (NanoGPT) with about 960 parameters. The model has a single transformer block with embedding dimension $n_{\mathrm{embd}} = 8$, one head of causal self-attention, vocabulary size 25, block size 16, and MLP ratio 1. The structure consists of:

- token and positional embeddings ($25 \times 8$ and $16 \times 8$),
- a causal self-attention layer (8 queries, keys, and values projected into 8 dimensions),
- a lightweight feedforward sublayer (linear $8 \to 8 \to 8$),
- final layer normalization and language modeling head ($8 \to 25$).

The resulting parameter count is approximately 960.

# E    DISCRETE SGD VS. LANGEVIN DYNAMICS: DETAILED COMPARISON

## E.1    INTRODUCTION

Below, we provide details of the similarities and differences between the discrete SGD dynamics and the continuous Langevin approach. The main statement is as follows:

**The Langevin approach does not reproduce the correct behavior of discrete SGD dynamics for some parameter values and finite learning rates $\eta$.**

Specifically, for certain directions in the parameter space (characterised by their own Hessian eigenvalues and variance), the Langevin approach predicts a finite parameter variance in the long-time limit. In contrast, it will grow exponentially if one considers the exact SGD behavior.

We demonstrate the origin of this incorrect prediction and the necessary corrections to address this inconsistency.

To clarify all the important points, we consider a simple one-dimensional example. Let us start from the exact SGD dynamics.

## E.2    DISCRETE SGD DYNAMICS

Consider the one step of the SGD dynamics

$$\theta_{n+1} = \theta_n - \eta \nabla L_n(\theta_n). \tag{46}$$

Here, $\eta$ denotes the learning rate, and $L_n(\theta_n)$ is the loss function that contains the noisy component. We assume stationarity and statistical independence of $L_n(\theta)$ at different steps. This random function can be characterized by its moments. The first two moments are given by:

$$\left\langle \nabla L_n(\theta) \right\rangle_L = \nabla \bar{L}(\theta), \qquad \left\langle \left( \nabla L_n(\theta) \right)^2 \right\rangle_L = D(\theta) + \left( \nabla \bar{L}(\theta) \right)^2. \tag{47}$$

In Appendix A of the main text, we prove that the probability density $P_n(\theta)$ satisfies the following exact recursion on the $n$-th step:

$$P_{n+1}(\theta) = \sum_{m=0}^{\infty} \frac{\eta^m}{m!} \nabla^m \Big( \left\langle \left( \nabla L(\theta) \right)^m \right\rangle_L P_n(\theta) \Big). \tag{48}$$

To second order in $\eta$ we obtain (see also (3) in the main text)

$$P_{n+1}(\theta) = P_n(\theta) + \eta\nabla\Big(\big(\nabla\bar{L}(\theta)\big)P_n(\theta)\Big) + \frac{1}{2}\eta^2\nabla^2\Big(\big(D(\theta) + \big(\nabla\bar{L}(\theta)\big)^2\big)P_n(\theta)\Big) + O(\eta^3). \tag{49}$$

Note that this "Fokker-Planck equation"-like relation exactly follows from the basic SGD step (46) without any assumptions. Now consider the Langevin equation approach.

### E.3 LANGEVIN EQUATION

The counterpart of the discrete SGD dynamics is a continuous-time model, which is given by Langevin equation (an Itô stochastic differential equation) (Li et al., 2019; Bach, 2022).

$$d\theta(t) = -\nabla\bar{L}(\theta(t))dt - \sigma(\theta(t))dW_t. \tag{50}$$

Usually the connection between the above equation and the SGD step (46) is established through the following steps:

1. Employment of the *Euler-Maruyama discretization*

$$\theta(t + \Delta t) = \theta(t) - \Delta t\nabla\bar{L}(\theta(t)) - \sqrt{\Delta t}\sigma(\theta(t))\Delta W_t, \tag{51}$$

   where $W_t$ is a Gaussian random variable with zero mean and unit variance.

2. Considering *timestep equal to the learning rate* $\Delta t = \eta$.

3. Identification of the *noise covariance* with random part of the loss $\sigma(\theta) = \sqrt{\eta D(\theta)}$.

4. Assumption of the *Gaussianity* of the noisy component of the loss function, which is essential for (50). It means that statistical properties of $\sigma(\theta)\Delta W_t$ and $\sqrt{\eta}(\nabla L_n(\theta) - \nabla\bar{L}(\theta))$ coincide (see (47)).

After that one can deduce that Euler-Maruyama discretization

$$\theta(t + \eta) = \theta(t) - \eta\nabla\bar{L}(\theta(t)) - \sqrt{\eta}\sigma(\theta(t))\Delta W_t \tag{52}$$

is the same as the

$$\theta(t + \eta) = \theta(t) - \eta\nabla\bar{L}(\theta(t)) - \eta(\nabla L_n(\theta) - \nabla\bar{L}(\theta)) = \theta(t) - \eta\nabla L_n(\theta(t)). \tag{53}$$

It means that one can model $\theta_n$ from the SGD (46) by the solution $\theta(t = n\eta)$ of the continuous-time Langevin equation

$$d\theta(t) = -\nabla\bar{L}(\theta(t))dt - \sqrt{\eta D(\theta)}dW_t. \tag{54}$$

From this equation one can derive the corresponding Fokker-Planck equation. It can be done with the help of the Euler-Maruyama discretization ((51) again. Up to the first order in $\Delta t$ one has

$$P(t + \Delta t, \theta) \approx P(t, \theta) + \Delta t\nabla\Big(\big(\nabla\bar{L}(\theta)\big)P(t, \theta)\Big) + \frac{1}{2}\eta\Delta t\nabla^2\Big(D(\theta)P(t, \theta)\Big) + O(\Delta t^2), \tag{55}$$

where $P(t, \theta)$ is the probability density at time $t$. In the $\Delta t \to 0$ limit one can reproduce the Fokker-Planck equation

$$\partial_t P(t, \theta) = \nabla\Big(\big(\nabla\bar{L}(\theta)\big)P(t, \theta)\Big) + \frac{1}{2}\eta\nabla^2\Big(D(\theta)P(t, \theta)\Big). \tag{56}$$

However, if we again set $\Delta t = \eta$ in the recursion (10), we will not reproduce the exact SGD recursion for probability density (49). This happens because we discarded the terms $\sim \Delta t^2 = \eta^2$ of the same order as the third contribution we left $\sim \eta\Delta t = \eta^2$. Only by taking it into account will we obtain the

correct answer that we derived earlier. This is the source of the inconsistency, and it can be corrected by considering the modified Langevin equation, namely

$$d\theta(t) = -\nabla\bar{L}(\theta(t))dt - \sqrt{\eta\big(D(\theta(t)) + \big(\nabla\bar{L}(\theta)\big)^2\big)}dW_t. \tag{57}$$

and the modified Fokker-Planck equation

$$\partial_t P(t,\theta) = \nabla\Big(\big(\nabla\bar{L}(\theta)\big)P(t,\theta)\Big) + \frac{1}{2}\eta\nabla^2\Big(\big(D(\theta) + \big(\nabla\bar{L}(\theta)\big)^2\big)P(t,\theta)\Big). \tag{58}$$

Thus we provide plausible arguments that (57), (58) is more suitable for the SGD description than (54), (56). However, in our main text we employ the exact (48) and do not use the continuous-time limit, because it might be the source of confusion.

In the next part we provide the simple illustrative example.

### E.4 SIMPLE MODEL

Let us consider the simple toy model of the SGD with a quadratic loss function

$$L(\theta) = L_0 + G\theta + \frac{1}{2}H\theta^2, \tag{59}$$

where $G$ and $H$ are the noisy gradient and the Hessian respectively. We assume that they are independent Gaussian random variables with moments

$$\langle G\rangle_L = 0, \quad \langle G^2\rangle_L = d, \quad \langle H\rangle_L = \lambda, \quad \langle H^2\rangle_L = \lambda^2 + \Gamma. \tag{60}$$

In this case

$$\nabla\bar{L}(\theta) = \lambda\theta, \qquad D(\theta) = d + \Gamma\theta^2. \tag{61}$$

Let us introduce the mean value of the parameter at the $n$-th time step

$$\mu_n = \int d\theta\,\theta P_n(\theta),$$

and its variance

$$\Pi_n = \int d\theta\,\theta^2 P_n(\theta) - \mu_n^2.$$

By multiplying the (50) by $\theta$ and $\theta^2$ and integrating it over $\theta$ one can derive the recursive relations for $\mu_n$, $\Pi_n$

$$\mu_{n+1} = \mu_n - \eta\lambda\mu_n = (1 - \eta\lambda)\mu_n. \tag{62}$$

$$\Pi_{n+1} = \Pi_n - 2\eta\lambda\Pi_n + \eta^2\lambda^2\Pi_n + \eta^2\Gamma\Pi_n + \eta^2\big(d + \Gamma\mu_n^2\big). \tag{63}$$

These relations are exactly the same as those derived in Appendix B of our paper. The solution has the form

$$\mu_n = (1 - \eta\lambda)^n\mu_0, \tag{64}$$

$$\Pi_n = \Big(\big((1-\eta\lambda)^2 + \eta^2\Gamma\big)^n - \big(1 - \eta\lambda\big)^{2n}\Big)\mu_0^2 + \eta^2 d\frac{1 - \big((1-\eta\lambda)^2 + \eta^2\Gamma\big)^n}{2\eta\lambda - \eta^2\lambda^2 - \eta^2\Gamma}. \tag{65}$$

Let us explore the long-time limit $n \to \infty$ of this solution. It depends strongly on the value of $1 - \eta\lambda$ and $(1 - \eta\lambda)^2 + \eta^2\Gamma$. In order for the mean value $\mu_n$ of the $\theta$ to converge for $n \to \infty$, it is necessary that $|1 - \eta\lambda| < 1$, or $0 < \eta\lambda < 2$. For the variance $\Pi_n$ the condition of long-time finiteness is $|(1 - \eta\lambda)^2 + \eta^2\Gamma| < 1$, or $\eta\Gamma < \lambda(2 - \eta\lambda)$. The first condition is related to the SGD convergence at finite $\eta$ and rather non-restrictive whereas the second one determines the desired condition. If conditions are satisfied, then in the long-time limit one has $\mu_{n\to\infty} = 0$ and

$$\Pi_{n\to\infty} = \frac{\eta d}{\lambda(2 - \eta\lambda) - \eta\Gamma}. \tag{66}$$

Now let us perform the same kind of analysis for the Langevin equation approach. As in the discrete case, let us introduce the mean value and the variance of the $\theta(t)$

$$\mu(t) = \int d\theta\,\theta P(t,\theta), \qquad \Pi(t) = \int d\theta\,\theta^2 P(t,\theta) - \mu^2(t).$$

The equations for these quantities can be derived from the Fokker–Planck equation (11), in a similar way to eq.(17) and (18)

$$\partial_t \mu(t) = -\lambda \mu(t), \tag{67}$$

$$\partial_t \Pi(t) = -2\lambda \Pi(t) + \eta \Gamma \Pi(t) + \eta \big(d + \Gamma \mu^2(t)\big). \tag{68}$$

From these equations it follows that the long-time behavior of $\mu(t)$ and $\Pi(t)$ is controlled by $\lambda$ and $2\lambda - \eta \Gamma$. The variance $\Pi(t)$ will be finite for $t \to \infty$ if $\eta \Gamma < 2\lambda$ and equal to.

$$\Pi(t \to \infty) = \frac{\eta d}{2\lambda - \eta \Gamma}. \tag{69}$$

This limiting value and condition $\eta \Gamma < 2\lambda$ differs from the exact condition $\eta \Gamma < \lambda(2 - \eta \lambda)$, which we derived for discrete SGD.

In contrast, if we perform the same steps for the modified Langevin (57) and Fokker-Planck (58) equations, then we obtain $\eta \Gamma < \lambda(2 - \eta \lambda)$ for finiteness criterion and

$$\Pi(t \to \infty) = \frac{\eta d}{\lambda(2 - \eta \lambda) - \eta \Gamma}. \tag{70}$$

for the limiting value.

Therefore, for $2\lambda - \eta \lambda^2 < \eta \Gamma < 2\lambda$ the widely used Langevin approach predicts the existence of stationary state with finite variance, whereas the exact discrete SGD dynamics shows that variance grows in time exponentially. Note that the modified Langevin (57) and Fokker-Planck (58) equations predict the same behavior as exact SGD.

## F   THE USE OF LARGE LANGUAGE MODELS (LLMS)

We use Large Language Models for text editing, i.e. grammar checking, word selection, text compression.

