# OpenReview forum: "Why SGD is not Brownian Motion: A New Perspective on Stochastic Dynamics"
_ICLR.cc/2026/Conference — Submitted to ICLR 2026_

### Official Review · Reviewer_BhVW · 2025-10-19

**Soundness:** 2
**Presentation:** 3
**Contribution:** 2
**Rating:** 2
**Confidence:** 4

**Summary:**

Different from previous studies, this paper argues that **SGD is best understood as deterministic dynamics within a fluctuating loss landscape** and models the parameter evolution with a Fokker-Planck equation. Under a quadratic assumption near local minima, the authors found that the parameters saturate along the positively curvatured directions, while diffusing along the near-flat directions with a rate propotional to the step size. This theorectial finding is further empirically validated on different tasks including CV and NLP.

**Strengths:**

This paper is well-written and easy to follow. The perspective of viewing SGD as a deterministic motion across a fluctuating loss landscape is interesting. Most importantly, the authors identified two distinct regimes near the local minima that the parameter evolution varies along different directions determined by the Hessian eigenvalues.

**Weaknesses:**

While this paper provides some useful insights on why SGD is not Brownian motion, there remains several issues on the novelty and the correctness of theoretical contribution:

 - First, the finding that flater directions correspond to smaller fluctuations has been extensively studied before (Feng & Tu, 2021a), for example, see [1][2][3]. Particuarly, [3] showed that the variance of stochastic gradient  is proportional to the magnitude of the corresponding eigenvalues of the Hessian. So the statement **this important observation challenged prevailing intuitions...** is not extactly true and the authors should change the wording approriately.

 [1] Chaudhari et al., Entropy-sgd: Biasing gradient descent into wide valleys, 2016.

 [2] Sagun et al., Empirical analysis of the hessian of over-parametrized neural networks, 2017.

 [3] Daneshmand et al., Escaping saddles with stochastic gradients, 2018.

 - Second, I was wondering the intrinsic difference between the two viewpoints: a random motion along static potential and a determinstic motion along random potential. The empirical loss is averged over the training set and each step we minimize it with a stochastic gradient, which of course the randomness is incurred by the mini-batching. As far as I can tell, the former viewpoint is more intuitive and reasonable. Particularly, [4] has already showed that SGD is running on a convoluted (smoothed) version of the loss function.

 [4] Kleinberg et al., An Alternative View: When Does SGD Escape Local Minima, 2018.

- Third, the assumptions are not verfied. Particularly, line 295 indicates the pre-dominant diagoality of $<H^2_{ij}>_{\delta L}$. But Figure 2(right) only demonstrates this for large variance index. Whether this still holds for small variance index?  I believe this is critically important for the derivation of the main result where $\lambda_i$ is close to zero. Moreover, I do not know how the constant $\epsilon$ appears in Equation (4). I notice that the authors introduce it at line 777, but how large it is when compared to $\lambda_i \approx 0$. My biggest concern is what will happen when $\lambda_i$ is much smaller than zero. We cannot preclude this occation, paticularly  when the training set is seriously unbalanced. Also, the used neural network is quite small, which may not reveal the true trend in the over-parameterized regime.

- Finally, it seems that the section of Conclusion is missing and some of the hyper-links are not properly formatted, e.g. line 197 "(3) provides..." and line 189 "Risken (1989)". Of course, some related works on modelling SGD with Fokker-Planck equation are missing as well, see [5][6].

[5] Tan et al., Understanding short-range memory effects in deep neural networks, 2023.

[6] Lucchi et al., On the Theoretical Properties of Noise Correlation in Stochastic Optimization, 2022.

**Questions:**

please see Weaknesses.

---

> ### Author Response · Authors · 2025-11-20
> **Part 1**
>
> Dear Reviewer,
> Thank you for your careful reading of our work and for the constructive feedback. Below, we respond to each of your concerns.
>
> **W1 (Prior work already shows that flatter directions correspond to smaller fluctuations; novelty of the observation is overstated)**
>
> **Response**: Thank you for this remark. We agree that our original phrasing was inappropriate. After revisiting the cited works, we removed the sentence in question and updated the text to ensure it does not overstate novelty.
>
> **W2 (Unclear conceptual distinction between “random motion on static potential” vs. “deterministic motion on random potential”; relation to existing interpretations such as smoothed loss landscapes)**
>
> **Response**: Thank you for raising this point — the distinction is indeed subtle, and we introduced it using physical intuition to clarify the conceptual difference between the two modeling approaches. In the Langevin framework, the dynamics correspond to a particle subject to random forces evolving in a static potential landscape. In contrast, SGD behaves as a deterministic update rule operating within a fluctuating potential, where the variability arises from minibatch-induced changes in the loss.
>
> This difference is well known in physics (e.g., Langevin diffusion vs. passive scalar advection), and we used this analogy to motivate why the two viewpoints need not be equivalent when the update rule is discrete and the noise is anisotropic.
>
> We have added new Appendix E to explain this distinction more clearly and to detail where the standard Langevin approximation may become inaccurate — in particular, due to the omission of discrete-time terms of order $\eta^2$ and the assumption of isotropic Gaussian noise. As illustrated in Appendix E, these differences can lead the two approaches to produce qualitatively different predictions in certain parameter regimes.
>
> Our intention was to provide intuitive motivation for why a discrete Fokker–Planck–type description may sometimes be more appropriate for SGD, and we hope the expanded clarification helps.

---

> ### Author Response · Authors · 2025-11-21
> **Part 2**
>
> **W3 (Assumptions not verified; diagonality only shown for large variance index; concerns about small-variance regime; unclear role and magnitude of constant γ; behavior when γ ≪ 0; dependence on small networks)**
>
> **Response**: Thank you for these detailed and insightful questions. We address each point below.
>
> 1. **Diagonality for small eigenvalues**. Figure 2(right) focuses on the large-variance regime simply because it is visually the most informative: only a small subset of Hessian eigenvalues is significantly non-zero (as seen in Figure 2(left)), and therefore only those directions yield a measurable covariance signal at finite sample sizes. For very small eigenvalues, the corresponding components of the SGD-induced covariance become extremely small as well. In this low-curvature regime the covariance matrix is still predominantly diagonal—we have verified this numerically—but the effect is difficult to display clearly on the same color scale without overwhelming the figure. We appreciate the reviewer’s remark and will revise the visualization to better highlight this regime.
> 2. **The constant ε in Equation (4)**. We introduced ε to ensure positivity of the denominator in cases where the Hessian has negative eigenvalues (which does occur early in training or on imbalanced datasets). Operationally, ε should be interpreted as a small positive regularization term whose scale is comparable to the magnitude of the most negative eigenvalue of the Hessian. Without this term, the expression in line 777 (in the previous version) would be inconsistent because the left-hand side is non-negative by definition. We also thank the reviewer for pointing out the missing parentheses in that equation — we have corrected this in the revised manuscript.
> 3. **Concern about small networks**. Regarding the concern about using a small neural network, our experimental choice is driven purely by feasibility: the quantities required to directly test our theory—full Hessian eigenstructure and minibatch fluctuation statistics—can only be computed exactly in small and medium-sized models. Larger modern architectures make such measurements computationally intractable.
>
> At the same time, this limitation is experimental, not theoretical. The curvature regime that our analysis relies on is known to persist in larger and over-parameterized networks. As shown in Feng & Tu (2021) [1], which we already cite, increasing network width does not alter the number of sharp directions, while the additional dimensions introduced by wider architectures remain flat. Their experiments in [1], including convolutional networks trained on MNIST and CIFAR-10, confirm that the sharp–flat separation central to our results remains stable as models scale.
>
> Thus, while our empirical tests use a tractable network size, the underlying geometric picture is supported by prior work and is expected to hold in the over-parameterized regime as well.
>
> **W4 (Missing conclusion section; formatting and hyperlink issues; missing related works on Fokker–Planck analyses)**
>
> **Response**: Thank you for pointing this out. In the revised manuscript we have:
>
> * Added a dedicated Conclusion section, summarizing our theoretical findings and their implications.
>
>
> * Corrected all formatting issues, including the misreferenced “(3) provides…” and the incorrectly formatted “Risken (1989)” citation.
>
>
> * Added the missing related works on Fokker–Planck-based analyses of SGD, specifically Tan et al. (2023) and Lucchi et al. (2022), and integrated them into the discussion of prior work to better situate our contribution in this literature.
>
>
> We believe these revisions substantially improve the polish, completeness, and contextual grounding of the manuscript.
>
> [1] Yu Feng and Yuhai Tu. The inverse variance–flatness relation in stochastic gradient descent is critical for finding flat minima. PNAS 118(9): e2015617118, 2021.
>
> **Q1 (See Weaknesses)**
>
> **Response**: We thank the reviewer for highlighting these issues. All concerns listed under the Weaknesses section have been addressed in the revised version:
>
> * the Conclusion section is now included;
> * formatting and hyperlink inconsistencies have been corrected;
> * and the discussion of related Fokker–Planck approaches has been expanded with additional references.

---

> ### Author Response · Authors · 2025-11-27
> **Follow-up on Rebuttal**
>
> Dear Reviewer BhVW,
>
> Thank you very much for the time and attention you devoted to reviewing our paper. We would like to send a brief follow-up regarding our rebuttal. We understand that your schedule is demanding, but we wanted to kindly check whether you have had a chance to consider our response.
>
> We believe that our clarification addresses the main concerns you raised, and we would be grateful for any additional comments or guidance you may wish to share.
>
> Best regards, The Authors

---

### Official Review · Reviewer_vnBz · 2025-10-24

**Soundness:** 3
**Presentation:** 1
**Contribution:** 2
**Rating:** 2
**Confidence:** 3

**Summary:**

This work investigates stochastic gradient descent (SGD) with resampling, where at every step the gradient is evaluated on a minibatch independently sampled from the training data with replacement. Its main contribution is to derive an approximation for the discrete Fokker-Planck equation for the evolution of the probability distribution of the weights in the small learning rate regime (Equation 2), capturing the leading order statistics of the SGD noise. This amounts to an interpretation of the stochastic SGD iterations as a deterministic evolution on a random potential.

The authors then investigate the behavior of the Fokker-Planck equation around a critical point of the expected loss, deriving a closed-form expression for the covariance matrix of the weights trajectory when expressed in the basis of the averaged Hessian (Equation 4), under a few assumptions on the Hessian (L290-L298).

The main take-aways from this characterization are that the variance along directions of positive curvature grows initially and then saturate and that nearly flat directions lead to a diffusive diffusive behavior.

Numerical experiments on Computer Vision and Natural Language processing are presented to illustrate the theoretical findings.

**Strengths:**

The main strength of the paper is that the setting studied is fairly general, with minimal assumptions made directly on the statistics of the loss rather than on the task. The insights drawn are interesting, and are corroborated by the numerical experiments.

**Weaknesses:**

The two main weaknesses of the manuscript are:

1. **Relation to previous work**. The first sentence in the abstract:
> "*The conventional wisdom in deep learning theory often models Stochastic Gradient Descent (SGD) as a Brownian particle, described by a Langevin equation. In this work, we challenge this paradigm [...]*"

is highly exaggerated and misleading, as it suggests that the study of the structure of SGD noise is largely unexplored. In reality, there exists an extensive body of literature in machine learning, known under the umbrella of "*SGD vs. SDEs*", that addresses precisely this question, yet the manuscript does not adequately relate its contributions to these prior works --- see for instance this [blog post](https://francisbach.com/rethinking-sgd-noise/) and the list of references therein. In particular, the result that to leading order in the learning rate only the covariance of minibatch fluctuations contributes to the dynamics overlaps with results from the line of work by (Li et al. 2017). I strongly suggest the authors revise the relation to previous work in a revised version, lowering the tone on the predominance of "SGD = Langevin" in the ML literature (e.g. L11-13, L42-44, L56-58) and better emphasizing their contribution to the extensive list of work that studied this question.

2. **The lack of polish and clarity**. The manuscript appears unpolished and gives the impression that it was submitted hastily, without adequate revision. The text would benefit from a thorough proofreading to correct the numerous typographical errors (a few examples are listed below, though the list is by no means exhaustive) and to improve the overall quality of the English (note that using LLMs for text correction is not prohibited by the [ICLR policy](https://blog.iclr.cc/2025/08/26/policies-on-large-language-model-usage-at-iclr-2026/) :) ) . Since the results strongly depend on the sampling scheme used, I suggest the authors define more explicitly SGD with replacement in the beginning of Section 3 for added clarity.

**References**:

- (Pillaud-Vivien 2022) Rethinking SGD's noise. URL: https://francisbach.com/rethinking-sgd-noise/
- (Li et al. 2017) Stochastic Modified Equations and Adaptive Stochastic Gradient Algorithms

**Typos**

- L196-197: "*This independence ensures that (3) provides an exact probabilistic description of SGD under this setting.*". What is (3) referring to here? Equation (3) only appears later in the text, and concerns the expansion of the loss around a critical point. Seems unrelated to this discussion.
- L281: "*Where $O_{ki}, O_{kl}$ is orthogonal matrix"
- L195: "*Within a such framework*"
- L280: "*Formally, eigenbasis of the*"
- L287-288 "*is almost diagonal matrix and small non-diagonal elements corresponds*"

**Questions:**

- Can the authors provide some intuition on what the assumptions 1-3 in L290-298 imply on the underlying problem? Do they have any concrete example where they can show these assumptions are satisfied? For example, what do they imply for a simple least-squares regression problem?

- In Figures 3 and 5, why the orders of magnitude of the y-axis in the theoretical (left) and numerical (right) plots differ? Minor note: the y-axix label in these plots seem to be cut.

---

> ### Author Response · Authors · 2025-11-20
>
> Dear Reviewer,
> Thank you for your detailed and constructive review of our work. Below, we respond to each of your comments.
>
> **W1 (Exaggerated and misleading positioning of the contribution; insufficient relation to extensive prior work on “SGD vs SDEs”)**
>
> **Response**: We have added a new Appendix E, where we clarify in detail how our approach relates to prior work on Langevin-based analyses of SGD. In that section, we explain the precise connection between our discrete Fokker–Planck formulation and existing SDE-based frameworks. We kindly invite the reviewer to take a look; if any part remains unclear, we will be happy to further refine the explanation.
>
> **W2 (Manuscript lacks polish and clarity; typos; unclear definitions; unclear references in the text; grammatical errors; missing details)**
>
> **Response**:  In the revised version we have improved the clarity of presentation: (i) we added an explicit definition of SGD with replacement at the beginning of Section 3; (ii) we introduced Appendix E, where we more carefully relate our results to prior work and expanded the list of references accordingly; (iii) we corrected errors mentioned in the review, fixed the inconsistent numbering of equations. We plan to perform a thorough proofreading pass to improve the overall readability of the manuscript. We believe these revisions significantly improve the clarity and polish of the paper.
>
> **Q1 (Intuition behind assumptions 1–3 in L290–298; concrete examples such as least-squares regression)**
>
> **Response**:
> Assumptions 1–3 are standard local assumptions in SGD theory, and each of them has a clear interpretation:
>
> * Assumption 1 is widely used in the literature — for example, it is the same assumption employed and empirically validated in Feng & Tu (2021) [1]. We also verified numerically that $H_{ij}$ components are mainly independent.
>
> * Assumption 2 is directly checked on our data.
>
> * Assumption 3 simply characterizes the regime where the Taylor expansion remains valid. In practice, the required condition still allows learning rates commonly used in deep learning, and is closely related to standard stability conditions for SGD (1/L).
> These assumptions are precisely the ones under which our theoretical derivation becomes valid.
> Thank you for the suggestion — checking the assumptions on simple linear regression is indeed a very good idea. We have not included this experiment in the current version, but we plan to add this verification in a future revision.
>
> **Q2 (Difference in orders of magnitude between theoretical and numerical plots in Figures 3 and 5; axis labels cut)**
>
> **Response**: Thank you for pointing this out. The mismatch in the absolute scale is expected: our theoretical expression predicts the variance only up to a multiplicative constant that does not depend on the learning rate. Because of this unknown prefactor, the theoretical and empirical curves need not coincide in magnitude. In Figures 3 and 5, our goal was to compare the qualitative behavior — growth, saturation, and diffusion regimes — and we verified that these trends match consistently across theory and experiment.
>
> Additionally:
>
> * We have added clarifying captions to Figures 3 and 5 explaining why the theoretical and numerical amplitudes differ;
> * Although the y-axis labels are currently cropped, we will correct this within the next few days.
>
>
> **References:**
>
> [1] Yu Feng and Yuhai Tu. The inverse variance–flatness relation in stochastic gradient descent is critical for finding flat minima. PNAS 118(9): e2015617118, 2021.
>
> We thank the reviewer once again for the thoughtful and constructive feedback. Your comments helped us significantly improve both the clarity and the positioning of the work. We hope that the revised version addresses all raised concerns.

---

> ### Author Response · Authors · 2025-11-27
> **Follow-up on Rebuttal**
>
> Dear Reviewer vnBz,
>
> Thank you very much for the time and attention you devoted to reviewing our paper. We would like to send a brief follow-up regarding our rebuttal. We understand that your schedule is demanding, but we wanted to kindly check whether you have had a chance to consider our response.
>
> We believe that our clarification addresses the main concerns you raised, and we would be grateful for any additional comments or guidance you may wish to share.
>
> Best regards, The Authors

---

### Official Review · Reviewer_7gYJ · 2025-10-31

**Soundness:** 3
**Presentation:** 3
**Contribution:** 3
**Rating:** 6
**Confidence:** 2

**Summary:**

The paper aims at improving the theoretical understanding of Stochastic Gradient Descent (SGD). It challenges the view that Stochastic Gradient Descent behaves like Brownian motion and is well-modeled by Langevin dynamics. Instead, it proposes that SGD is better understood as deterministic dynamics in a fluctuating loss landscape, with stochasticity arising from mini-batch sampling rather than external noise, leading to potentially non-gaussian gradient noise.

In particular, paper mains contributions are:
1) Derivation of a master equation and a discrete Fokker–Planck equation for SGD, highlighting differences from the standard Langevin approach.
2) Analysis Near Critical Points: Detailed study of SGD dynamics near minima, using the Hessian eigenbasis to distinguish between "rigid" (high curvature) and "diffusive" (flat) directions.
3) Experiments on vision (MNIST MLP) and language (NanoGPT on Shakespeare) tasks confirm theoretical predictions, especially the separation into diffusive and rigid modes.

**Strengths:**

- Paper is well motivated The limitations of the Langevin approach are clearly articulated, especially regarding noise scaling.

- The paper provides a novel and rigorous theoretical framework for understanding SGD, moving beyond the Langevin/Brownian analogy.

- It rigorously derives the Fokker–Planck equation for SGD, clarifying the role of mini-batch noise.

- The analysis in the Hessian eigenbasis is insightful, allowing for a clear separation of parameter dynamics. The connection between gradient and Hessian fluctuations is well-motivated and supported by derivations.

**Weaknesses:**

- The empirical validation is restricted to small models and datasets (e.g., 2 layers MLP on MNIST, NanoGPT on Tiny Shakespeare). It is unclear how well the theory scales to large, modern architectures and datasets.

- The analysis relies on several assumptions (e.g., independence of Hessian fluctuations in particular). It would be nice to expand on the trade-off and accuracy of those assumptions..

- While the theoretical insights are strong, the paper does not discuss in detail how these findings could influence practical optimization strategies.

**Questions:**

- How do the theoretical predictions hold for large-scale models (e.g., deep CNNs, Transformers) and real-world datasets? Are there empirical results or intuitions for such settings?

- What happens if the independence or diagonality assumptions about the Hessian are violated? Can the theory supports such as case?

---

> ### Author Response · Authors · 2025-11-20
>
> Dear Reviewer,
> Thank you for your thoughtful and constructive review of our work. We appreciate your positive assessment of the motivation and theory, and we address your concerns point by point below.
>
> **W1 (Empirical validation limited to small models and datasets)**
>
> **Response**: Thank you for raising this point. Our choice of experimental scale is driven by a practical limitation: the key quantities required to validate our theory — the Hessian eigenstructure and the minibatch-induced covariance — can be computed exactly only for small and medium-sized models. For modern architectures, full Hessian analysis is computationally infeasible, so smaller models allow us to verify our theoretical predictions in a controlled and fully observable regime.
> Importantly, the curvature structure underlying our analysis is not specific to small models. As shown in Feng & Tu (2021) [1], which we already cite in the manuscript, the qualitative Hessian geometry remains stable as network size increases. In particular, they write:
> “As the network size (width H) increases, the number of sharp directions does not change, and the landscapes along the additional degrees of freedom are flat with large values of flatness.”
> Moreover, they verify this behavior on larger architectures, including convolutional networks trained on MNIST and CIFAR-10. This provides independent evidence that the sharp-vs-flat separation — central to our analysis — persists in larger, modern models as well.
>
> **W2 (Reliance on assumptions, in particular independence/diagonality of Hessian fluctuations; need to discuss trade-offs and accuracy)**
>
> **Response**: All assumptions used in our analysis are either empirically verified or validated indirectly by demonstrating that the simplified theoretical model still matches the experimental results. In cases where the assumptions are approximations (e.g., independence of Hessian fluctuations), we show that relaxing them would make the theory intractable, yet the simplified version already reproduces the qualitative and quantitative behavior observed in our experiments.
>
> **W3 (Lack of discussion of practical implications for optimization strategies)**
>
> **Response**: Our work is primarily theoretical in nature, aiming to clarify the fundamental mechanisms underlying SGD dynamics. For this reason, we do not focus on practical optimization strategies, although our findings may inspire future work in that direction.
>
> **Q1 (Scalability of theoretical predictions to large-scale models and real-world datasets)**
>
> **Response**: While a full eigenbasis analysis is prohibitively expensive for modern architectures, the theoretical mechanism we derive is architecture-agnostic: it depends only on the local curvature structure and minibatch-induced fluctuations. As discussed above, prior work such as Feng & Tu (2021) [1] demonstrates that the sharp–flat structure of the Hessian persists even as network width grows and across larger architectures (e.g., CNNs on CIFAR-10). Since our predictions hinge precisely on this anisotropic curvature regime, the same saturation–diffusion behavior is expected to hold in deep CNNs, Transformers, and large-scale datasets. A clarification addressing this issue has been added to Section 5 in the revised manuscript.
>
> **Q2 (Behavior of the theory when independence/diagonality assumptions on the Hessian are violated)**
>
> **Response**: We note that the full theory does not require independence or diagonality. A more general formulation—based on superoperators acting on the full space of covariance matrices—can handle fully coupled Hessian fluctuations. However, we chose to present the simplified version in the paper because it already matches the experimental data well enough, as illustrated in Figures 3 and 5. Introducing the full superoperator formalism would considerably complicate the exposition without changing the empirical conclusions.
>
> **References:**
> [1] Yu Feng and Yuhai Tu. The inverse variance–flatness relation in stochastic gradient descent is critical for finding flat minima. PNAS 118(9): e2015617118, 2021.
>
> We thank the reviewer once again for the thoughtful and constructive feedback. The clarifications and revisions introduced in the updated manuscript address the raised concerns and improve both the clarity and overall strength of the paper.

---

> ### Author Response · Authors · 2025-11-27
> **Follow-up on Rebuttal**
>
> Dear Reviewer 7gYJ,
>
> Thank you very much for the time and attention you devoted to reviewing our paper. We would like to send a brief follow-up regarding our rebuttal. We understand that your schedule is demanding, but we wanted to kindly check whether you have had a chance to consider our response.
>
> We believe that our clarification addresses the main concerns you raised, and we would be grateful for any additional comments or guidance you may wish to share.
>
> Best regards,
> The Authors

---

### Official Review · Reviewer_5zFC · 2025-11-01

**Soundness:** 2
**Presentation:** 2
**Contribution:** 2
**Rating:** 4
**Confidence:** 4

**Summary:**

This is a paper tackling an interesting problem: insightful characterisation of SGD in terms of analysable continuous time analogues. It first describes the problem in terms of a fluctuating gradient field, considers the Fokker-plank equation, derives some perturbation analysis for the diffusion in the vicinity of a critical point, and does a number of related experiments.

**Strengths:**

This paper considers an important issue, and provides a direction of analysis, via a perturbation analysis around a critical point to obtain the dynamics at that critical point, and define the covariance of the diffusion dynamics around any critical point. It derives a potential useful analysis and does experiments on key models corresponding to that analysis. I commend the researchers for their work and encourage them to push it forward, but for some of the reasons below, I do not think this paper is ready for airtime just yet.

The paper would benefit from:

A clear take-home. Why is there a benefit in doing all this analysis? Why does that help use.

A precise distinction between this analysis and one based on Lengevin diffusions, if that exists, starting with the FP and demonstrating explicitly that the analysis of SGD cannot be viewed as _any_ general form of diffusion, and that this distinction has actual impact on understanding.

Alternatively, remove the claim of a distinction, and work hard at relating the work to other diffusion analyses of SGD. Try removing invalid assumptions or make explicit that they are invalid, but insight can still be made by making them.

Clean the paper writing up, define the problem being addressed, the insight that the authors bring to the paper and the point of the paper: why does the paper change the view of the world?

Tighten up the experiments to address the precise conclusions and do controlled ablations that make those conclusions convincing.

**Weaknesses:**

There are a few undefined terms on unexplained equations in parts. Explain the meaning of different indices etc.

"Nevertheless, this construction does not faithfully reproduce the actual SGD update, in which the stochastic contribution
naturally enters at order η, but not √η."

I think this is not correct: There are two parameters in SGD: learning rate and batch size, whose setting relate to one another, and between them they determine the relationship between the noise and drift terms in the corresponding Euler-Maruyama discretised diffusion. The stochasticity in SGD can then be controlled to be of whatever order one desires (see for example https://arxiv.org/abs/1711.04623 - already referenced). Furthermore the Lengevin formalism as discussed says nothing about the covariance structure of \xi, which is critical to the formalism.

The statement that SGD is "deterministic dynamics in a fluctuating potential." is not insight but a statement of the obvious: it is what the SGD equation just says. Each new minibatch defines a new potential, related to one another as they are determined samples from the underlying full-batch. The insight for considering the Lengevin analogue comes from the recognition that the derivative of these potentials are unbiased (and independent if constructed properly) samples from some distribution, considered to be the noise distribution.

This paper is therefore suggesting that this direction is a red-herring and we should return to the basic starting viewpoint that "SGD is deterministic dynamics in a fluctuating potential" and arguably take a different tack. The question is then whether this claim is justified, or indeed whether this different tack is a different tack at all, or just the same thing written in a different way. I find this paper is just going back to the base and rederiving the original Lengevin formalism, but with the relevant covariance structure handled. However most papers in this field consider the problem of this covariance structure and there has been a long history of papers working out how best to characterise it, estimate it, or model it.

(as an aside, please number all equations. How am I as a reviewer to communicate about equations if there are no numbers to refer to?)

In fact by considering the Fokker-Planck equation, the structure is inherently viewing this in terms of diffusion, it can end up being exactly the same as the Lengevin formalism, with the Hessian of the noise relating to the covariance structure of the noise terms. Most Lengevin analyses of learning dynamics understand this in terms of the Fokker-Planck equation. See https://proceedings.mlr.press/v97/zhu19e/zhu19e.pdf (already referenced) for example, and https://papers.nips.cc/paper_files/paper/2015/file/6f4922f45568161a8cdf4ad2299f6d23-Paper.pdf. So I am feeling that so far this is standard fare dressed up as something new, and I am not convinced this is outside of a Lengevin understanding of the problem.

Then on to the analysis of the near-critical point. This is specific to this paper.

In the derivation the "simplifying assumptions, which are rather natural for NN under consideration. Namely we assume
that fluctuations of all components of ˜Gi and ˜Hij are independent." is not at all fair. By all means make that assumption, but considering them natural for neural networks is not accurate; given the impact of each parameter on the output is the product of all parameters along the unblocked paths from input to output, these are heavily non-independent.

It is not really helpful to state that ˜Πnij is predominantly diagonal" when you have assumed (without validity) that H_{ij} is diagonal, given it dominates. Technically it is correct given the assumptions, but gives the wrong impression to the reader.

How does this result compare with previous related analsyses? I would want this to be set in the scene of the extensive existing work in this field. In particular, one paper of Zhanxing Zhu's group is referenced, but his other work considers many of these issues. Broader discussion would be very valuable.

I am really unsure about the take-home here. There is no conclusion, and the observations that there are rigid and diffusive directions is longstanding and a result of the overall loss surface and not of this formalism. The same results come out of previous Lengevin analysis. I fear I am a bit lost as to what conclusion I should draw from this paper.

I think writing this as "we move away from Lengevin to fluctuating gradient fields and then heading back to look at the Fokker-Planck equation is a bit of a circle, given the general formalism of Lengevin methods. Given most of the analysis is critical point analysis, there is actually no difference between a Lengevin and non-Lengevin diffusion analysis at such a point. If you really want to push this point I would start with a general diffusion formalism, the Fokker Planck and show exactly that there is no potential for which this is a Lengevin diffusion of any sort, and then demonstrate that this observation buys you something very significant. I don't see it at the moment, I am afraid.

**Questions:**

What conclusions do you want me to draw from this? What is the fundamental understanding I now have as a result of your paper that will change my view of the world?

What am I missing about your argument?

---

> ### Author Response · Authors · 2025-11-20
> **Part 1**
>
> Dear Reviewer,
> We are grateful for your detailed feedback and for the time and effort you have invested in reviewing our paper. Below, we address each of the concerns you have raised.
>
> **W1 (Langevin vs discrete SGD approach  )**
>
> **Reviewer comment:** Why is there a benefit in doing all this analysis? Why does the paper change the view of the world?
>
> **Response:** We have added an additional section in Appendix E, where we demonstrate the distinction between the discrete SGD and the Langevin dynamics approach. We provide a one-dimensional toy example with explicit calculations, showing a scenario where this difference affects the observable behavior.
> Please refer to our explanations in the newly added Appendix E. If you find these clarifications valuable, we will include them in the main body of the paper.
>
> **W2 (Independence of components of $G_i$ and $H_{ij}$ )**
>
> **Reviewer comment:** Namely we assume that fluctuations of all components of ˜Gi and ˜Hij are independent." is not at all fair. By all means make that assumption, but considering them natural for neural networks is not accurate; given the impact of each parameter on the output is the product of all parameters along the unblocked paths from input to output, these are heavily non-independent.
>
> **Response:** Yes, you are absolutely correct that assuming the independence of the H and G components is not universally justified for all neural networks. We developed a general theory of dynamics near a critical point; however, obtaining a fully general closed-form solution is not feasible.
> Therefore, we introduced the assumptions described in our work. The resulting theoretical predictions align well with the experimental data, at least for MLP and NanoGPT, as shown in Figures 3 and 5 in the main text.
> Thus, we have demonstrated that even such a simplified theory captures the essential properties of the dynamics near a critical point.
>
> **W3 (Deterministic dynamics in a fluctuating potential)**
>
> **Reviewer comment:** The statement that SGD is "deterministic dynamics in a fluctuating potential." is not insight but a statement of the obvious: it is what the SGD equation just says.
>
> **Response:** We used physical intuition to highlight the difference between the Langevin equation approach and discrete SGD. The Langevin equation originally describes the dynamics of a particle moving under a random force in a static potential. In contrast, as you rightly noted, SGD dynamics correspond rather to the motion of a deterministic particle in a fluctuating potential—which, in physics, corresponds to the advection of a passive scalar. The methods for describing such dynamics differ from the Langevin equation approach. Although the two descriptions are very similar for SGD, as we showed in the supplementary text, there exist parameter regimes where these two methods yield different results. We wish to clarify that for SGD, which is inherently discrete, it might not be necessary to rely on the Langevin equation, and it could be preferable to work with a discrete equation resembling the Fokker-Planck equation.
>
> **W4 (Diagonality of $Π_n^{ij}$ )**
>
> **Reviewer comment:** It is not really helpful to state that $Π_n^{ij}$ is predominantly diagonal" when you have assumed (without validity) that $H_{ij}$ is diagonal, given it dominates
>
> **Response:** We do not assume that $H_{ij}$ is diagonal. Instead, we transition into the eigenbasis where this matrix is diagonal and perform our analysis in this basis. We empirically verify that $Π_n^{ij}$ remains primarily diagonal in this representation. In principle, there might exist neural network architectures where $Π_n^{ij}$ would not be diagonal in such a basis. This would substantially complicate the theoretical analysis. A more general treatment without this assumption is substantially more involved and is left for future work.

---

> ### Author Response · Authors · 2025-11-20
> **Part 2**
>
> **W5 (Comparison with literature)**
>
> **Reviewer comment:** I would want this to be set in the scene of the extensive existing work in this field. In particular, one paper of Zhanxing Zhu's group is referenced,
>
> **Response:** Appendix E partially addresses this point and offers a comparison with the widely used approach also found in the works you referenced. Should our explanation in this text remain insufficient, we will certainly provide a more detailed clarification.
>
> **W6 (There is no conclusion)**
>
> **Reviewer comment:** There is no conclusion
>
> **Response:** We agree with the reviewer on the importance of a clear conclusion and have therefore added a dedicated section to summarize our findings.
>
> **W7 (Unclear what fundamental understanding the reader should gain; reviewer is unsure what to conclude)**
>
> **Reviewer comment:** What conclusions do you want me to draw from this? What is the fundamental understanding I now have as a result of your paper that will change my view of the world?
>
> **Response:** The key message we aim to convey is twofold:
> The Langevin equation approach is not the only viable framework, and for describing the discrete dynamics of SGD, an alternative methodology is likely more appropriate.
> We demonstrate that a Fokker-Planck-type equation, derived directly from the discrete SGD dynamics and combined with our analysis near a critical point, provides a sufficiently accurate description of the phenomena observed in NN experiments.
>
> Thank you again for your thoughtful and constructive review.

---

> ### Author Response · Authors · 2025-11-27
> **Follow-up on Rebuttal**
>
> Dear Reviewer 5zFC,
>
> Thank you very much for the time and attention you devoted to reviewing our paper. We would like to send a brief follow-up regarding our rebuttal. We understand that your schedule is demanding, but we wanted to kindly check whether you have had a chance to consider our response.
>
> We believe that our clarification addresses the main concerns you raised, and we would be grateful for any additional comments or guidance you may wish to share.
>
> Best regards,
> The Authors

---

### Author Response · Authors · 2025-11-21
**Rebuttal Revision**

We thank all reviewers for their insightful and constructive feedback. Below we summarize the main revisions introduced in the updated manuscript:
1) Added a new Appendix E clarifying in detail how our discrete Fokker–Planck formulation relates to prior Langevin/SDE-based analyses of SGD, including a discussion of when the two viewpoints diverge.


2) Revised the introduction and abstract to avoid overstating novelty regarding fluctuation–curvature relationships, and removed the inaccurate sentence flagged by the reviewers.


3) Expanded the discussion of assumptions (Lines 290–298), with clearer intuition, justification, and explanation of empirical verification; added notes on the small-curvature regime and clarified the role of the constant ε.


4) Improved clarity in Section 3, including an explicit definition of “SGD with replacement.”


5) Corrected all identified formatting issues, including equation numbering, misreferenced hyperlinks, and citation formatting (e.g., “(3) provides…” and “Risken (1989)”).


6) Added missing related work, including recent Fokker–Planck-based analyses (Tan et al., 2023; Lucchi et al., 2022), and expanded the contextualization of our contribution within the broader “SGD vs. SDEs” literature.


7) Added a Conclusion section summarizing the core findings and their implications.


8) Improved Figures 3 and 5, adding clarifying captions explaining the difference in scale between theory and experiment; noted that y-axis labels will be corrected in the final revision.


9) Performed a broad proofreading pass, fixing the typos listed by reviewers and improving overall clarity; a full polishing pass will be completed before the camera-ready version.


10) We believe these revisions substantially improve the clarity, correctness, and positioning of the manuscript, and we are grateful to the reviewers for helping us strengthen the paper.

---

### Meta-Review · Area_Chair_6C79 · 2025-12-28

**Summary:**

The paper contributes to the ongoing debate on modeling SGD as SDE in practical training of neural networks. The paper starts by emphasizing the disconnect between modeling SGD as brownian motion, in particular in terms of the empirical anisotropy (aligned with curvature) and heavy tailed nature of the gradients.  The paper proposes a novel theoretical model that makes a novel assumption on the (mini-batch) fluctuation of the loss-surface and derives the corresponding Fokker-Planck equation, which is then analytically studied near the critical point through a set of simplifying assumptions.

Reviewers appreciated sound theory and the overall picture of deterministic dynamics happening in a fluctuating loss landscape.

The paper also does a great job in describing complex mathematical modeling without obscuring the message with technical aspects.

Reviewers had however critical feedback that couldn’t have been fully addressed during the discussion phase. Reviewers vnBz and 5zFC raised concern about limited engagment with existing SGD vs SDE literature, including comment that the paper presents misleading framing of “SGD ≈ Langevin” as dominant wisdom.

On the AC side, the paper should more directly engage with works on progressive sharpening/break-even point, and more directly try to model the early phase of training (vs the critical point). For example, Cohen et al recently studied a theoretical model that’s consistent with progressive sharpening. Relatedly, progressive shaprening seems to clash with the assumption that learning rate is small compared to the Hesisan. Potentially that’s resolved by the fact it is the “mean” Hessian, but is not clear.

All in all, the paper presents interesting theory, but should more directly compare to existing theoretical models of SGD, especially ones (in the opinion of AC) that directly model progressive sharpening.

**Reviewer Concerns:**

Reviewers were not clear on the Assumptions made. Reviewers added experiments in the discussion phase that largely addressed this issue.

Reviewers vnBz and 5zFC raised a critical concern about limited engagment with existing SGD vs SDE literature. In particular, Reviewer vnBz raised concern of the paper misleading framing of "SGD ≈ Langevin” as dominant wisdom. This has been partially addressed in the rebuttal, but for example there is no discussion of progressive sharpening and reaching break-even point made in the paper, which are essential phenomena in gradient-based training of neural networks that directly related to the “SGD vs SDE” debate.

**Reviewer Scores:**

Reviewers might have increased scores based on some of the rebuttal (e.g. the accuracy of assumptions), but would have been unlikely fully convinced regarding positioning of the work in the broader debate of how to model SGD.

---

### Decision · Program_Chairs · 2026-01-26

Reject